# ONLINE EPSILON NET & PIERCING SET FOR GEOMETRIC CONCEPTS[*]

**Sujoy Bhore**[†]  **Devdan Dey**[‡]  **Satyam Singh**[§]

## ABSTRACT

*VC-dimension* (Vapnik & Chervonenkis (1971)) and *ε-nets* (Haussler & Welzl (1987)) are key concepts in Statistical Learning Theory. Intuitively, *VC-dimension* is a measure of the size of a class of sets. The famous *ε-net theorem*, a fundamental result in Discrete Geometry, asserts that if the VC-dimension of a set system is bounded, then a small sample exists that intersects all sufficiently large sets.

In online learning scenarios where data arrives sequentially, the VC-dimension helps to bound the complexity of the set system, and *ε*-nets ensure the selection of a small representative set. This sampling framework is crucial in various domains, including spatial data analysis, motion planning in dynamic environments, optimization of sensor networks, and feature extraction in computer vision, among others. Motivated by these applications, we study the *online ε-net* problem for geometric concepts with bounded VC-dimension. While the offline version of this problem has been extensively studied, surprisingly, there are no known theoretical results for the online version to date. We present the first deterministic online algorithm with an optimal competitive ratio for intervals in $\mathbb{R}$. Next, we give a randomized online algorithm with a near-optimal competitive ratio for axis-aligned boxes in $\mathbb{R}^d$, for $d \leq 3$. Furthermore, we introduce a novel technique to analyze similar-sized objects of constant description complexity in $\mathbb{R}^d$, which may be of independent interest.

Next, we focus on the continuous version of this problem (called *online piercing set*), where ranges of the set system are geometric concepts in $\mathbb{R}^d$ arriving in an online manner, but the universe is the entire ambient space, and the objective is to choose a small sample that intersects all the ranges. Although *online piercing set* is a very well-studied problem in the literature, to our surprise, very few works have addressed generic geometric concepts without any assumption about the sizes. We advance this field by proposing asymptotically optimal competitive deterministic algorithms for boxes and ellipsoids in $\mathbb{R}^d$, for any $d \in \mathbb{N}$.

## 1 INTRODUCTION

The concepts of Vapnik–Chervonenkis dimension (VC-dimension) and $\varepsilon$-net theory are fundamental components in Statistical Learning Theory. VC-dimension, introduced by Vapnik & Chervonenkis in their seminal work Vapnik & Chervonenkis (1971), is a tighter measure of the complexity of concept classes. We need some key definitions to discuss the notion of VC-dimension formally. A set system (also known as *range space*) $(\mathcal{X}, \mathcal{R})$ is defined by a set $\mathcal{X}$ and a class $\mathcal{R}$ (known as ranges) of subsets of $\mathcal{X}$. For instance, consider a set system: $\mathcal{X} = \{1, 2, 3, 4\}$ and $\mathcal{R} = \{\{1, 2\}, \{2, 3\}, \{2, 3, 4\}, \{1, 2, 4\}\}$. (see also Figure 1 for a geometric example). In learning theory, the set $\mathcal{X}$ is the instance space, and $\mathcal{R}$ is the class of potential hypotheses, where a hypothesis $r$ is a subset of $\mathcal{X}$. A set system $(\mathcal{X}, \mathcal{R})$ *shatters* a set $\mathcal{A}$ if each subset of $\mathcal{A}$ can be expressed as $\mathcal{A} \cap r$ for some $r$ in $\mathcal{R}$. The VC-dimension of $\mathcal{R}$ is the size of the largest set shattered by $\mathcal{R}$. Due to Vapnik & Chervonenkis (1971), it is known that for any range space $(\mathcal{X}, \mathcal{R})$ with VC-dimension bounded by a constant $d$, for any $\varepsilon > 0$, a randomly chosen *small* subset of $\mathcal{X}$ will *hit* every range containing at least $\varepsilon|\mathcal{X}|$ points from $\mathcal{X}$, with high probability. Haussler & Welzl (1987) showed that the size of the *small* subset, called an $\varepsilon$-net (for formal definition, see Definition 1 in Section 2), is bounded by $O\left(\frac{d}{\varepsilon} \log \frac{d}{\varepsilon}\right)$, where $d$ is the VC-dimension of the range space. This result is famously

---

[*]All authors contributed equally.

[†]Department of Computer Science and C-MInDS, Indian Institute of Technology Bombay, Mumbai, India. Email: `sujoy.bhore@gmail.com`.

[‡]Department of Computer Science & Engineering, Indian Institute of Technology Bombay, Mumbai, India. Email: `mnz.devdan@gmail.com`.

[§]Department of Computer Science & Engineering, Indian Institute of Technology Bombay, Mumbai, India. Email: `satyamiitd19@gmail.com`.

known as the $\varepsilon$-*net theorem*, and is a celebrated result in Discrete Geometry. One of the central open questions in the theory of $\varepsilon$-nets is whether the logarithmic factor $\log \frac{1}{\varepsilon}$ in the upper bound on their size is truly necessary. Pach & Woeginger (1990) showed for $d \geq 2$, logarithmic factor is necessary, but $d = 1$ the net size can be bounded by $\max\left(2, \left\lfloor \frac{1}{\varepsilon} \right\rfloor - 1\right)$. In the last three decades, remarkable progress has been made on the size of $\varepsilon$-net for *geometric set families* by exploiting various intrinsic geometric properties (we briefly discuss these results in Section 1.1).

In this work, we focus on the $\varepsilon$-net problem in the online setup. In the *online $\varepsilon$-net* problem, the set $\mathcal{X}$ is known in advance, but the objects of $\mathcal{R}$ arrive one at a time, without advance knowledge, and we need to maintain a valid net $N \subset \mathcal{X}$ for the input objects. The performance of an *online $\varepsilon$-net* algorithm is measured by the *competitive ratio*, which is (informally) defined as the maximum ratio between the performance of the algorithm and the offline optimal net (see Section 2 for a formal definition).

Besides its underlying deep theoretical nature, *online $\varepsilon$-nets* have found many applications in modern machine learning, particularly in areas like active learning, adversarial robustness, efficient sampling, etc. In active learning, $\varepsilon$-nets help to select representative samples from large datasets. This process allows the models to be trained with minimal labelled data while maintaining accuracy (see, e.g., Hanneke & Yang (2015); Balcan et al. (2010)). Moreover, *online $\varepsilon$-nets* play an important role in adversarial robustness by covering potential adversarial regions of the input space, ensuring that models are less susceptible to attacks (see, e.g., Madry et al. (2017); Cullina et al. (2018); Montasser et al. (2019)). In this work, we primarily focus on the theoretical aspects of *online $\varepsilon$-nets*, which form a crucial component of Statistical Learning Theory, contributing to our understanding of generalization, sample complexity, and robustness in machine learning models (Vapnik (2013); Laan et al. (2006)).

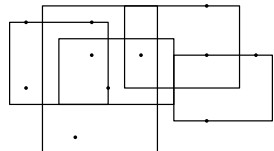

Figure 1: An example of a geometric range space $(\mathcal{X}, \mathcal{R})$. Here, the set $\mathcal{X}$ is the collection of points, while $\mathcal{R}$ is the collection of rectangles.

**Continuous Setup: Towards Piercing Set.** Given a set $\mathcal{R}$ of $n$ geometric objects in $\mathbb{R}^d$, a subset $P \subset \mathbb{R}^d$ is a *piercing set* of $\mathcal{R}$ if every object of $\mathcal{R}$ contains at least one point of $P$. The *minimum piercing set* (MPS) problem asks for a piercing set $P$ of the smallest size. The problem has numerous applications in facility location, wireless sensor networks, learning theory, etc. See Sharir & Welzl (1996); Huang et al. (2004); Ben-Moshe et al. (2000); Katz et al. (2003); Mustafa (2022). The problem can be viewed as a "continuous" version of $\frac{1}{n}$-net problem, also known as the geometric hitting set problem. The geometric hitting set, in turn, corresponds to *geometric set cover* in the dual range space[1]. Hence, by the standard greedy algorithm for set cover, one can compute an $O(\log n)$-approximation to the minimum piercing set in polynomial time for any family of piercing set with constant description complexity (since it suffices to work with a discrete set of $O(n^d)$ candidate points). For *geometric set families*, a range of sophisticated approximation schemes have been proposed over the years (see Section 1.1 for a brief discussion).

In *online piercing set*, the point set $\mathbb{R}^d$ is known beforehand, but the set $\mathcal{R}$ of geometric objects is not known in advance. Here, the geometric objects arrive one by one. An *online algorithm* must maintain a valid piercing set for all objects arrived so far. Upon the arrival of a new object $\sigma$, the algorithm must maintain a valid piercing set. Note that an online algorithm may add points to the piercing set but cannot remove points from it, i.e., all the decisions taken by the algorithm are irrevocable. The problem aims to minimize the cardinality of the piercing set. In the *online hitting set*, $\mathcal{X} \subset \mathbb{R}^d$ such that $|\mathcal{X}| = n$. Charikar et al. (2004) initiated the study of the online piercing set problem for unit balls in $\mathbb{R}^d$. They proposed an online algorithm having a competitive ratio of $O(2^d d \log d)$. Moreover, they proved that $\Omega(\log d / \log \log \log d)$ is the (deterministic) lower bound of the competitive ratio for this problem. Later, Dumitrescu et al. (2020) improved both the

---

[1]Given a finite family $\mathcal{R}$ of ranges in $\mathbb{R}^d$, the dual range space induced by them is defined as a set system on the underlying set $\mathcal{R}$, consisting of the sets $\mathcal{R}_x := \{R \mid x \in R \in \mathcal{R}\}$, for all $x \in \mathbb{R}^d$.

upper and lower bounds of the competitive ratio to $O(1.321^d)$ and $\Omega(d + 1)$, respectively. For unit hypercube in $\mathbb{R}^d$, Dumitrescu & Tóth (2022) proved that the competitive ratio of any deterministic online algorithm for the unit covering problem is at least $2^d$. Then, for integer hypercubes in $\mathbb{R}^d$, they proposed a randomized online algorithm with a competitive ratio of $O(d^2)$ and a deterministic lower bound of $d + 1$. For similar size $\alpha$-fat objects in $\mathbb{R}^d$, De et al. (2024a) gave a deterministic algorithm with competitive ratio $O((\frac{2}{\alpha} + 2)^d \log M)$, and a lower bound of $\Omega(d \log M + 2^d)$. Note that a set $\mathcal{S}$ is said to be *similarly sized $\alpha$-fat objects* when the ratio of the largest width of an $\alpha$-fat object (for the definition of $\alpha$-fat object, see Section 2) in $\mathcal{S}$ to the smallest width of an $\alpha$-fat object in $\mathcal{S}$ is bounded by a fixed constant $M > 0$. See Section 1.1 for further discussion on this.

An *online $\varepsilon$-net* can be viewed as a specific type of *online piercing set* where the focus is on maintaining coverage with respect to the measure of the sets rather than merely ensuring intersection. Consequently, both structures aim to address the complexities of dynamic data scenarios by providing robust sampling and representation mechanisms.

## 1.1 RELATED WORK

**$\varepsilon$-net:** The $\varepsilon$-net theory has seen remarkable growth in the last few decades. Here, we provide a very concise summary of this. Matoušek (1992) demonstrated that for range spaces $(\mathcal{X}, \mathcal{R})$ where $\mathcal{X}$ is a finite set of points in $\mathbb{R}^2$ (or $\mathbb{R}^3$) and $\mathcal{R}$ consists of half-spaces, the size of the $\varepsilon$-net can be reduced to $O((\frac{1}{\varepsilon})$ eliminating the logarithmic factor. Aronov et al. (2009) showed the existence of $\varepsilon$-nets of size $O\left(\frac{1}{\varepsilon} \log \log \frac{1}{\varepsilon}\right)$ for planar point sets and axis-aligned rectangles. Clarkson & Varadarajan (2005) made an important breakthrough by establishing a connection between the size of $\varepsilon$-nets for dual range space $(\mathcal{X}, \mathcal{R})$ associated with geometric objects and their *union complexity*[2]. In particular, they showed if the union complexity is $o(n \log n)$, then dual set systems admit $\varepsilon$-net of size $o(1/\varepsilon \log(1/\varepsilon))$. On the lower bound side, one can typically find approximately $1/\varepsilon$ pairwise disjoint, $\varepsilon$-heavy ranges in $\mathcal{R}$. For these cases, the size of any $\varepsilon$-net must be at least $\Omega\left(\frac{1}{\varepsilon}\right)$. For many years, it has been widely conjectured that for *geometric set families*, this bound is tight (see Matoušek et al. (1990)). Alon (2012) proved the conjecture false by giving examples of geometric range spaces of small VC-dimension, e.g., straight lines, rectangles, or infinite strips in the plane, that do not have $\varepsilon$-net of size $O(1/\varepsilon)$. Later, Pach & Tardos (2011) showed that range spaces with VC-dimension 2 have a smallest $\varepsilon$-net of size $\Omega(1/\varepsilon \log 1/\varepsilon)$. They also proved lower bound on size of $\varepsilon$-net for axis-parallel rectangle in $\mathbb{R}^2$ is $\Omega(1/\varepsilon \log \log 1/\varepsilon)$.

**Piercing Set.** In the offline setting, the piercing set problem is a well-studied problem in Computational Geometry. The problem is NP-complete even for a set of unit squares Garey & Johnson (1979). For geometric set families, e.g., unit squares/hypercubes, unit disks/balls, or more generally, near-equal-sized fat objects in $\mathbb{R}^d$, various approximation schemes have been developed; see Chan (2003); Efrat et al. (2000); Katz et al. (2003)). For arbitrary boxes in $\mathbb{R}^d$, the current best approximation scheme is via *$\varepsilon$-net* (see Agarwal et al. (2024)). Recently, Bhore & Chan (2025) obtained a dramatic improvement over the running time.

*Online Piercing & Hitting.* Alon et al. (2009) in their seminal work initiated the study of the hitting set problem in the online setting. They proposed an online algorithm having a competitive ratio of $O(\log n \log m)$, where $|\mathcal{X}| = n$ and $|\mathcal{R}| = m$. Moreover, they establish a nearly matching $\Omega\left(\frac{\log m \log n}{\log \log m + \log \log n}\right)$ lower bound for the problem. In the geometric setting, Even & Smorodinsky (2014) proposed online algorithms having an optimal competitive ratio of $\Theta(\log n)$, where $\mathcal{X}$ is a finite subset of points and $\mathcal{R}$ consists of half-planes in $\mathbb{R}^2$, and also when $\mathcal{R}$ consists of unit disks. Khan et al. (2023) obtained an optimal $\Theta(\log N)$-competitive algorithm when $\mathcal{X}$ is a finite set of points from $\mathbb{Z}^2$ and $\mathcal{R}$ consists of integer squares (whose vertices have integral coordinates) $S \subseteq [0, N)^2$ in $\mathbb{R}^2$. Recently, De et al. (2024b) also obtained an optimal competitive ratio of $\Theta(\log n)$ when $\mathcal{X}$ is a finite set of points from $\mathbb{R}^2$ and $\mathcal{R}$ consists of translates of either a disk or a regular $k$-gon. For a special case, when the point set is entire $\mathbb{Z}^d$, De & Singh (2024) studied the problem for unit balls and hypercubes in $\mathbb{R}^d$. Alefkhani et al. (2023) considered this variant for $\alpha$-fat objects in $(0, N)^d$, and proposed a deterministic online algorithm with a competitive ratio of $(\frac{4}{\alpha} + 1)^{2d} \log N$. Recently, De et al. (2024b) obtained improved upper and lower bounds.

---

[2] The complexity of the boundary of the union of a set of objects (see Clarkson & Varadarajan (2007)).

## 1.2 OUR CONTRIBUTIONS.

We study the *online $\varepsilon$-net* and *online piercing set* for a wide range of geometric objects. For some of the objects, we designed online algorithms which achieve asymptotically tight competitive ratios. We summarize our results below.

**Online $\varepsilon$-net.** We present the first deterministic online algorithm for intervals in $\mathbb{R}$ with an optimal competitive ratio of $\Theta(\log \frac{1}{\varepsilon})$. This result is tight, as we also establish a lower bound of $\Omega(\log \frac{1}{\epsilon})$ for online $\varepsilon$-net for intervals. Next, for axis-aligned rectangles in $\mathbb{R}^2$ and boxes in $\mathbb{R}^3$, we devise randomized algorithms with near-optimal competitive ratios of $O(\log \frac{1}{\varepsilon})$ and $O(\log^3 \frac{1}{\varepsilon})$, respectively. We make significant progress on classical $\varepsilon$-net in the online regime, for which no prior upper bounds were known.

**Online piercing set.** Starting from the work of Charikar et al. (2004), *online piercing set* has been studied extensively over the years (see Dumitrescu et al. (2020); Dumitrescu & Tóth (2022)). However, it is impossible to obtain sublinear competitive ratios for any geometric families due to a hopeless lower bound of $\Omega(n)$, which even holds for arbitrary intervals, where $n$ is the length of the input sequence. Several works addressed this issue by making assumptions on the object types (*fatness*) or aspect ratio of the input objects (see De et al. (2024a); Khan et al. (2023)). Surprisingly, very little is known when these constraints do not hold. We present the first deterministic online algorithm for axis-aligned boxes and ellipsoids in $\mathbb{R}^d$, with an optimal competitive ratio of $O(\log M)$. These results are asymptotically tight due to the existing lower bound of $\Omega(\log M)$ for hypercubes and balls in $\mathbb{R}^d$ (De et al. (2024b)). Additionally, we introduce a novel technique to analyze similar-sized fat objects of constant description complexity in $\mathbb{R}^d$. Although the result slightly improves the existing upper bound of De et al. (2024a), we believe the technique may be useful to other online geometric algorithms. Due to paucity, we move the proofs of several lemmas, theorems, and pseudo-codes in the Appendix. The missing proofs are marked by $\star$.

## 2 NOTATION AND PRELIMINARIES

We use $\mathbb{Z}^+$ and $\mathbb{R}^+$ to denote the set of positive integers and positive real numbers, respectively. We use $[n]$ to represent the set $\{1, 2, \ldots, n\}$, where $n \in \mathbb{Z}^+$. For any $\beta \in \mathbb{R}$, we use $\beta\mathbb{Z}$ to denote the set $\{\beta z \mid z \in \mathbb{Z}\}$, where $\mathbb{Z}$ is the set of integers. For any point $p \in \mathbb{R}^d$, we use $p(x_i)$ to denote the $i$th coordinate of $p$, where $i \in [d]$. The point $p$ is an *integer point* if for each $i \in [d]$, the coordinate $p(x_i)$ is an integer. By an *object*, we refer to a compact set in $\mathbb{R}^d$ having a nonempty interior. Let $d_\infty(.,.)$ represents the distance under the $L_\infty$-norm. Given a set system $(\mathcal{X}, \mathcal{R})$ and any set $Y \subseteq \mathcal{X}$, the *projection* of $\mathcal{R}$ onto $Y$ is defined as the set system: $\mathcal{R}|_Y = \{Y \cap r : r \in \mathcal{R}\}$. The *VC-dimension* of $\mathcal{R}$, denoted by VC-dim($\mathcal{R}$) is the size of the largest $Y \subseteq \mathcal{X}$ for which $\mathcal{R}|_Y = 2^Y$.

**Definition 1.** *($\varepsilon$-net) Given a set system $(\mathcal{X}, \mathcal{R})$ (also known as range space) consists of a finite set $\mathcal{X}$ and a class $\mathcal{R}$ of subsets of $\mathcal{X}$. An $\varepsilon$-net $N \subset \mathcal{X}$ such that any range $r \in \mathcal{R}$ with $|r \cap \mathcal{X}| \geq \varepsilon \cdot |\mathcal{X}|$ intersects $N$. In other words, any range that has at least a proportion $\varepsilon$ of the elements of $P$ must also intersect the $\varepsilon$-net $N$.*

**Theorem 1.** *(Epsilon-net Theorem) [Haussler & Welzl (1987)] Let $(\mathcal{X}, \mathcal{R})$ be a set system with VC-dim($\mathcal{R}$) $\leq d$ for some constant $d$, and let $\varepsilon > 0$ be a given parameter. Then there exists an absolute constant $c_a > 0$ such that a random $N$ constructed by picking of $X$ independently with probability $\left(c_a.(\frac{1}{\varepsilon|X|}) \log \frac{1}{\gamma} + \frac{d}{\varepsilon|X|} \log \left(\frac{1}{\varepsilon}\right)\right)$ is an $\varepsilon$-net for $\mathcal{R}$ with probability at least $1 - \gamma$.*

**Online $\varepsilon$-Net.** Let $\Sigma = (\mathcal{X}, \mathcal{R})$ be a set system, where $\mathcal{X}$ is a universe of points in $\mathbb{R}^d$ and $\mathcal{R}$ is a set of ranges defined over $\mathcal{X}$. We assume that $\mathcal{X}$ is fixed in advance and the ranges in $\mathcal{R}$ are coming one by one. Let ALG be an algorithm for *online $\varepsilon$-net* for $\Sigma$. The expected competitive ratio of ALG with respect to $\Sigma = (\mathcal{X}, \mathcal{R})$ is defined by, $\rho(\text{ALG}) = \sup_\sigma \left[\frac{\text{ALG}(\sigma)}{\text{OPT}(\sigma)}\right]$, where the supremum is taken over all input sequences $\sigma$, OPT($\sigma$) is the minimum cardinality $\varepsilon$-net for $\sigma$, and ALG($\sigma$) denotes the size of the net produced by ALG for this input. The objective is to design an algorithm that obtains the minimum competitive ratio. If the ALG is a randomized algorithm, then we replace ALG($\sigma$) by $\mathbb{E}[\text{ALG}(\sigma)]$ (Borodin & El-Yaniv, 1998, Ch. 1).

$\alpha$-**Fat Objects.** The notions of *fatness* have been heavily exploited in high-dimensional Geometry and Learning Theory. There exists several definitions of fatness in the literature due to Alefkhani et al. (2023); Chan (2003); De et al. (2024a). We use the most standard definition here which is defined with respect to the aspect ratio of the objects. Let $\sigma$ be an object. For any point $x \in \sigma$, we define $\alpha(x) = \frac{\min_{y \in \partial\sigma} d_\infty(x,y)}{\max_{y \in \partial\sigma} d_\infty(x,y)}$. The *aspect ratio* $\alpha(\sigma) = \max\{\alpha(x) : x \in \sigma\}$. An object is

considered an *α-fat object* if its aspect ratio is at least $\alpha$. A point $c \in \sigma$ with $\alpha(c) = \alpha(\sigma)$ is defined as a *center* of the object $\sigma$. The minimum (respectively, maximum) distance from the center to the boundary of the object is referred to as the *width* (respectively, *height*) of the object. A set $\mathcal{S}$ of objects is considered *fat* if there exists a constant $0 < \alpha \leq 1$ such that each object in $\mathcal{S}$ is $\alpha$-fat. Note that for a set $\mathcal{S}$ of fat objects, each object $\sigma \in \mathcal{S}$ does not need to be convex or connected. For an $\alpha$-fat object, the value of $\alpha$ is invariant under translation, reflection, and scaling. A set $\mathcal{S}$ is said to be *similarly sized fat objects* when the ratio of the largest width of an object in $\mathcal{S}$ to the smallest width of an object in $\mathcal{S}$ is bounded by a fixed constant.

## 3 ONLINE $\varepsilon$-NET

In Section 3.1, we analyze the performance of a simple deterministic algorithm for online $\varepsilon$-net of arbitrary intervals, which gives asymptotically tight competitive ratios. Then, in Section 3.2, we analyse the performance of a randomized algorithm for online $\varepsilon$-net of arbitrary boxes in $\mathbb{R}^d (d \leq 3)$.

### 3.1 ONLINE $\varepsilon$-NET FOR ARBITRARY INTERVALS

In this section, we consider a finite range space $(\mathcal{X}, \mathcal{R})$, where $\mathcal{X}$ is a set of $n$ points in $\mathbb{R}$, and $\mathcal{R}$ is the set of arbitrary intervals. In the online setting, the set $\mathcal{X}$ is known in advance, and the adversary introduces the intervals one by one at each step. Our objective is to construct an $\varepsilon$-net $\mathcal{N} \subset \mathcal{X}$ for $(\mathcal{X}, \mathcal{R})$ that hits each $\varepsilon$-heavy interval in $\mathcal{R}$.

We present a deterministic online algorithm ALGO-INTERVAL, which maintains an $\varepsilon$-net $\mathcal{N}$. Initially, $\mathcal{N} = \emptyset$. At each step, we update the set to hit all the $\varepsilon$-heavy intervals observed so far. Here, OPT refers to the optimal $\varepsilon$-net produced by an offline algorithm that computes the best possible solution.

Let $\sigma$ be an interval containing $n$ points. We partition the interval $\sigma$ into 2 disjoint smaller sub-intervals, each containing at most $\lfloor n/2 \rfloor$ points. Let $P_\sigma^j$ be a $j$th sub-interval of $\sigma$, where $j \in \{\ell, r\}$.
**Online algorithm.** We can now present our online algorithm $ALG$. The algorithm maintains a piercing set $\mathcal{H}$ for all intervals that have been part of the input so far. Initially, $\mathcal{H} = \emptyset$. Upon the arrival of a new interval $\sigma$, if $|\sigma \cap \mathcal{X}| < \varepsilon|\mathcal{X}|$, then ignore $\sigma$; else we do the following. If it is already hit by $\mathcal{H}$, ignore $\sigma$. Otherwise, sort the points of $\sigma \cap \mathcal{X}$ in the increasing order, say $p_1, \ldots, p_{|\sigma \cap \mathcal{X}|}$, and hit $\sigma$ by the point indexed $\left\lfloor \frac{|\sigma \cap \mathcal{X}|}{2} \right\rfloor$ and $\left\lceil \frac{|\sigma \cap \mathcal{X}|}{2} \right\rceil$. Add the above-mentioned points points to $\mathcal{H}$. For a concise description of pseudo-code, see Algorithm 1 in Appendix C.

Since we hit all the $\varepsilon$-heavy unhit sets at each step, clearly, ALGO-INTERVAL produces an $\varepsilon$-net. We need to prove that the size of the net $\mathcal{N}$ produced by ALGO-INTERVAL is at most $2 \left( \log \left( \frac{1}{\varepsilon} \right) + 1 \right)$ times the size of the offline optimal net OPT.

**Theorem 2** ($\star$). *For online $\varepsilon$-net of arbitrary intervals, there exists a deterministic online algorithm with a competitive ratio of $2 \left( \log \left( \frac{1}{\varepsilon} \right) + 1 \right)$, for any $\varepsilon \in (0, 1]$. This result is tight: the competitive ratio of any deterministic online algorithm for this problem is at least $\log \frac{1}{\varepsilon} + 1$.*

### 3.2 ONLINE $\varepsilon$-NET FOR ARBITRARY AXIS-ALIGNED RECTANGLES IN $\mathbb{R}^2$

In this section, we consider a finite range space $(\mathcal{X}, \mathcal{R})$, where $\mathcal{X}$ is a set of $n$ points in $\mathbb{R}^2$, and $\mathcal{R}$ is the set of axis-aligned rectangles. In the online setup, the points are known in advance, and the rectangles are introduced one by one. Our goal is to construct an $\varepsilon$-net $N$ for $(\mathcal{X}, \mathcal{R})$ that hit all $\varepsilon$-heavy rectangles in $\mathcal{R}$. Before describing the online algorithm, we first introduce some crucial ingredients that will play an essential role in designing the algorithm.
**Construction of the balanced binary tree $\mathcal{T}$.** We construct a balanced binary search tree $\mathcal{T}$ over the point set $\mathcal{X}$, where $|\mathcal{X}| = n$. Without loss of generality, assume $n = 2^k$ for some $k \in \mathbb{Z}^+$. The root node at level 0 contains all $n$ points. At each level, the parent node splits into two child nodes, each containing half of the points of its parent. This process continues until each node has fewer than $\varepsilon n$ points, resulting in a tree of depth $O \left( \log \left( \frac{1}{\varepsilon} \right) \right)$, with each leaf containing $O(\varepsilon n)$ points.

**Construction of a random sample.** Let $\mathcal{P}$ be a random sample of size $O \left( \varepsilon \log \log \left( \frac{1}{\varepsilon} \right) \right)$, where $\varepsilon \in \left( \frac{1}{C}, 1 \right]$ for sufficiently large constant $C > 1$. The points are drawn uniformly at random from $\mathcal{X}$ with a probability of $\pi = O \left( \frac{\varepsilon \log \log \left( \frac{1}{\varepsilon} \right)}{n} \right)$. Note that the selection of $\mathcal{P}$ and its size is extremely crucial, as it directly influences the competitive ratio of the algorithm.
**Connection between the tree and the random sample.** Each node $v$ of the tree $\mathcal{T}$ is associated with a subset $\mathcal{X}_v \subseteq \mathcal{X}$ (and similarly, $\mathcal{P}_v \subseteq \mathcal{P}$) containing the points of $\mathcal{X}$ (resp. $\mathcal{P}$) stored in the

subtree rooted at $v$. A line $l_v$ corresponding to each internal node $v$ divides the point set $\mathcal{X}_v$ into two subsets, $\mathcal{X}_{v_1}$ and $\mathcal{X}_{v_2}$, associated with the children $v_1$ and $v_2$ of $v$, respectively. Corresponding to each node $v_i$ (except root) and it's parent $v$, lines $l_{v_i}$ and $l_v$ define the strip $s_{v_i}$.

**Construction of maximal $\mathcal{P}_v$-unhit open rectangles set $\mathcal{M}_v$.** For each node $v$ in strip $s_v$, containing $|\mathcal{P}_v|$ points from $\mathcal{P}$, located between lines $l_v$ and $l_{\text{parent}(v)}$. Without loss of generality, let $l_v$ be the left boundary of $s_v$. We need at most three points from $\mathcal{P}_v$ to create a $\mathcal{P}$-unhit open rectangle $M$. A triplet of points $(a, b, c)$ defines three sides of a rectangle $M$: right ($a$), top ($b$), and bottom ($c$), with the left side determined by the line $l_v$ (refer figure 2a). If either $b$ or $c$ is missing, the corresponding side is extended infinitely (see Figures 2b). If $a$ is missing, the right side of $M$ extends until it reaches the right boundary line of the strip $s_v$ (see Figure 2c). For each point $a \in \mathcal{P}_v$, the nearest top-left ($b$) and bottom-left ($c$) neighbors define the upper and lower boundaries of the rectangle. If no such neighbors exist, those sides are extended to infinity. This process generates up to $|\mathcal{P}_v|$ rectangles, where each right side is defined by a point from $\mathcal{P}_v$ (Figure 2d). Rectangles defined solely by the top and/or bottom points ($b$ and/or $c$) are formed by pairing consecutive points along the y-axis, with the right side extending to the opposite boundary of the strip (Figure 2e). This leads to the construction of $|\mathcal{P}_v| + 1$ rectangles. Thus, each strip $s_v$ contains up to $2|\mathcal{P}_v| + 1$ maximal $\mathcal{P}$-unhit rectangles.

From the above description, the number of maximal $\mathcal{P}_v$-unhit open rectangles $\mathcal{M}_v$ within each strip $s$ is bounded by $O(|\mathcal{P}|)$. Since the number of nodes in the tree $\mathcal{T}$ is $O(2^{\log \frac{1}{\varepsilon}})$, the total number of maximal $\mathcal{P}$-unhit open rectangles across all strips is at most $O\left(|\mathcal{P}| \cdot \frac{1}{\varepsilon}\right)$.

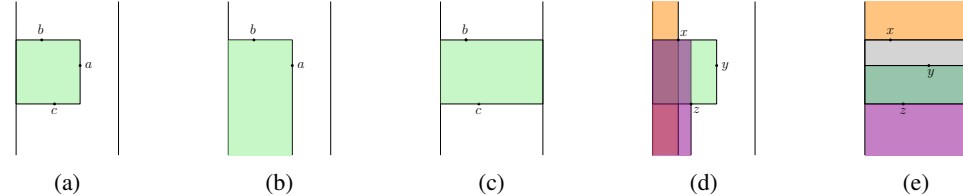

(a)    (b)    (c)    (d)    (e)

Figure 2: (a) A triplet of points $(a, b, c)$ defines three sides of a rectangle $M$. (b) If either *top* or *bottom* point is missing, the corresponding side is extended infinitely. (c) If *right* point is missing, the right side of $M$ extends until it reaches the right boundary line of the strip. (d) Rectangles defined solely by *right* side point. (e) Rectangles defined solely by *top* and/or *bottom* side point.

**Finding a suitable $\mathcal{P}_v$-unhit open rectangle.** If an $\varepsilon$-heavy rectangle $\sigma$ arrives and is not hit by $\mathcal{P}$, let $v$ be the highest node of $\mathcal{T}$ such that associated line $l_v$ intersects $\sigma$. We take the sub-rectangle $\sigma'$, which contains at least $\frac{\varepsilon n}{2}$ points. Next, we extend $\sigma'$ to the right until it hits a point of $\mathcal{P}_v$ or reaches the opposite boundary of the strip $s_v$. Similarly, we extend it upwards (resp., downwards) until it intersects a point of $\mathcal{P}_v$, or treat it as an open rectangle. This extended rectangle contains $\sigma'$ and is included in the set $\mathcal{M}_v$.

**Construction of safety-net.** For each node $v$ of $\mathcal{T}$ and each rectangle $M \in \mathcal{M}_v$, define the weight as $w_M = \frac{s|M \cap \mathcal{X}|}{n}$, where $s = \frac{2}{\varepsilon}\delta$, and $\delta$ is a small constant greater than 1. Using the $\varepsilon$-net theorem, we can construct a $\frac{1}{w_M}$-net, denoted as $N_M$, for each $M \cap \mathcal{X}_v$, of size $O(w_M \log w_M)$. These serve as *safety-nets* that hit every $\varepsilon$-heavy input rectangle $\sigma$ that intersects the strip $s_v$ but is not hit by $\mathcal{P}$.

The final $\varepsilon$-net $N$ for $(\mathcal{X}, \mathcal{R})$ is the union of $\mathcal{P}$ with the safety-nets $N_M$ for all $M$. Now, we have all the necessary ingredients to describe the algorithm.

**Online algorithm.** Let $\mathcal{P} \subseteq \mathcal{X}$ be a random sample of size $O\left(\varepsilon \log \log \left(\frac{1}{\varepsilon}\right)\right)$. In addition to $\mathcal{P}$, the algorithm also maintains a safety-net $SN$, with $N = \mathcal{P} \cup SN$. Initially, $\mathcal{I}$, $\mathcal{P}$, and $SN$ are empty. For each new rectangle $\sigma$ presented, update $\mathcal{I} = \mathcal{I} \cup \sigma$. If $\sigma$ contains any point from $\mathcal{P}$, then we are done. If not, check whether $\sigma$ intersects any point from the current safety-net $SN$. If it does, no further action is needed. Otherwise, find the highest node $v$ in the tree $\mathcal{T}$ where the associated line $l_v$ intersects $\sigma$. Identify a sub-rectangle $\sigma' \subseteq \sigma$ containing at least $\frac{\varepsilon n}{2}$ points, then extend $\sigma'$ to form a $\mathcal{P}_v$-unhit rectangle $M$. Finally, add all points from the $\frac{1}{w_M}$-net $N_M$ to the safety-net $SN$. For a concise description of the pseudo-code, see Algorithm 2 in Appendix C.

Readers familiar with the technique of Aronov et al. (2009) will recognize the similarities between our approach and theirs. However, the key distinctions in our algorithm lie in a different selection of the random sample, a different weight assigned to each constructed rectangle $M \in \mathcal{M}$, and consequently, the size of the resulting online $\varepsilon$-net changes.

**Correctness.** Note that $N$ consists of a random sample $\mathcal{P} \subseteq \mathcal{X}$ along with $w_M$-net for each $M \in \mathcal{M}$. Let $\sigma$ be a $\varepsilon$-heavy rectangle. If the input rectangle $\sigma$ is hit by $\mathcal{P}$, we are done. If $\sigma$ does not contains any point from $\mathcal{P}$, then we will show that $\sigma$ will be hit by a point from the safety-net $N_M$, corresponding to some $\mathcal{P}$-unhit open rectangle $M$. Recall that if an $\varepsilon$-heavy rectangle $\sigma$ arrives and is not hit by $\mathcal{P}$, we consider the highest node $v$ of $\mathcal{T}$ such that the associated line $l_v$ intersects $\sigma$. We take the sub-rectangle $\sigma'$, which contains more than half points of $\sigma$. Next, we find the maximal $P_v$-unhit open rectangle $M \in \mathcal{M}_v$ such that $\sigma'$ is completely contained in $M$. Also, recall that the weight of every $M$ was $w_M = \frac{s|M \cap \mathcal{X}|}{n}$, where $s = \frac{2\delta}{\varepsilon}$. Due to $\varepsilon$-net theorem, for any $M \in \mathcal{M}_v$ we can construct $1/w_M$-net, $N_M$ for $M$. Note that $\frac{|\sigma' \cap \mathcal{X}|}{|M \cap \mathcal{X}|} \geq \frac{\varepsilon n/2}{n w_M/s} \geq \frac{1}{w_M}$. Hence, due to the definition of $\varepsilon$-net that $N_M$ hits $\sigma'$.

**Theorem 3.** *For the online $\varepsilon$-net problem with arbitrary axis-aligned rectangles, there exists an algorithm with an expected competitive ratio of at most $O\left(\log\left(\frac{1}{\varepsilon}\right)\right)$. Here, $\varepsilon \in \left(\frac{1}{C}, 1\right]$, where $C$ is a sufficiently large constant.*

*Proof.* Now, we will show that the expected competitive ratio of the algorithm is $O\left(\log\left(\frac{1}{\varepsilon}\right)\right)$. Let $\mathcal{I}$ be the collection of all input rectangles arrived one by one to the algorithm. First, we will compute how many points are placed by our algorithm for the input sequence $\mathcal{I}$. Let $N$ and $\mathsf{OPT}$ be the $epsilon$-net constructed by our online algorithm and best offline optimum for input sequence $\mathcal{I}$. Recall that the net $N$ constructed by our algorithm is the union of $\mathcal{P}$ and $w_M$-net for all $M \in \mathcal{M}$, where $\mathcal{M}$ is the collection of all $\mathcal{P}$-unit open rectangles. Now, we consider

$$\mathbb{E}[|N|] = \mathbb{E}[|\mathcal{P}| + \sum_{v \in \mathcal{T}} \sum_{M \in \mathcal{M}_v} (w_M \log w_M)] \leq \mathbb{E}[|\mathcal{P}'|] + \mathbb{E}[|\mathcal{M}|(w_M \log w_M)]$$

$$= \mathbb{E}[|\mathcal{P}'|] + (w_M \log w_M)\mathbb{E}[|\mathcal{M}|] \leq O\left(\left(\frac{\mathbb{E}[|\mathcal{P}|]}{\varepsilon}\right) w_M \log w_M\right)$$
$$\text{(Since, } \mathbb{E}[|\mathcal{M}|] \text{ dominates over } \mathbb{E}[|\mathcal{P}'|])$$

$$= O\left(\log\log\left(\frac{1}{\varepsilon}\right)\right) \times (w_M \log w_M) = O\left(\log\log\left(\frac{1}{\varepsilon}\right)\right) \times O\left(\left(\frac{1}{\varepsilon}\right)\log\left(\frac{1}{\varepsilon}\right)\right).$$
$$\text{(Since, } w_M = O(s) \text{ and } s = \frac{2\delta}{\varepsilon})$$

Due to Pach & Tardos (2011), for $(\mathcal{X}, \mathcal{R})$, where $\mathcal{X}$ is a finite set of points, and $\mathcal{R}$ consists of axis-aligned rectangles, the size of the smallest $\varepsilon$-net (for $\varepsilon \in (0, 1]$) is at least $\Omega(\frac{1}{\varepsilon}\log\log\frac{1}{\varepsilon})$. Thus, the offline optimal $\mathsf{OPT}$ for any input sequence $\mathcal{I}$ will have size at least $O((\frac{1}{\varepsilon}\log\log\frac{1}{\varepsilon}))$. So, the expected competitive ratio of the algorithm will be $\frac{\mathbb{E}[|N|]}{\mathsf{OPT}} \leq \frac{O\left(\log\log\left(\frac{1}{\varepsilon}\right) \times O\left(\left(\frac{1}{\varepsilon}\right)\log\left(\frac{1}{\varepsilon}\right)\right)\right)}{O\left(\left(\frac{1}{\varepsilon}\right)\log\log\left(\frac{1}{\varepsilon}\right)\right)} = O\left(\log\left(\frac{1}{\varepsilon}\right)\right)$. Hence, the theorem follows. $\square$

Since we are using the Pach & Tardos (2011) result as a lower bound, the claimed upper bounds are not instance-optimal. Achieving instance-optimal bounds would require an online lower bound for these objects. To the best of our knowledge, such a result has not yet been established in the literature, making it an intriguing open problem.

### 3.2.1 EXTENSION TO HIGHER DIMENSIONS.

Our approach can be extended from $\mathbb{R}^2$ to $\mathbb{R}^3$. We begin by selecting a random sample $\mathcal{P} \subseteq \mathcal{X}$ in $\mathbb{R}^3$ of size $O\left(\varepsilon \log\log\frac{1}{\varepsilon}\right)$, similar to the case in $\mathbb{R}^2$. Then, we construct a three-level range tree $\mathcal{T}$ over the points of $\mathcal{X}$ using standard methods from Computational Geometry (see Berg et al. (2008)). This three-level range tree will help in construction of at most $O\left(|\mathcal{P}| \cdot \frac{1}{\varepsilon}\log^2\frac{1}{\varepsilon}\right)$ many safety-nets. (For the complete construction see Appendix A.2). Thus, we have the following theorem.

**Theorem 4** ($\star$)**.** *For the online $\varepsilon$-net problem with arbitrary axis-aligned boxes in $\mathbb{R}^3$, there exists an algorithm with an expected competitive ratio of at most $O\left(\log^3\left(\frac{1}{\varepsilon}\right)\right)$. Here, $\varepsilon \in \left(\frac{1}{C}, 1\right]$, where $C$ is a sufficiently large constant.*

**Remark 1.** *For dimensions $d \geq 4$, the number of maximal $\mathcal{P}$-unhit open orthants within each octant containing $k$ points from $\mathcal{P}$ may no longer be linear in $k$. In fact, it can grow as $\Theta(k^{\lfloor d/2 \rfloor})$, which is at least quadratic for $d \geq 4$ (see Kaplan et al. (2008)). This contrasts with instances in $\mathbb{R}^2$ or $\mathbb{R}^3$, where the number of such maximal unhit boxes is linear in $k$, allowing us to efficiently bound the net size, which results in a small net size and a favorable competitive ratio. However, due to the potentially non-linear growth in higher dimensions, it is unclear whether the tree construction algorithm used to find a small $\varepsilon$-net will yield similarly efficient results for $d \geq 4$.*

# 4 ONLINE PIERCING SET PROBLEM

In this section, we study the *online piercing set* problem for various families of geometric objects. In Section 4.1 and 4.2, we analyze the performance of a simple deterministic algorithm ALGO-CENTER for piercing axis-aligned boxes and ellipsoids in $\mathbb{R}^d$, respectively, which gives the desired competitive ratios. Then, in Section 4.3, we analyse the performance of a deterministic algorithm ALGO-FAT for piercing fat objects in $\mathbb{R}^d$.

In what follows, we first describe a simple deterministic algorithm.

**Online algorithm: ALGO-CENTER.** *Let $\mathcal{N}$ be the piercing set maintained by our algorithm to pierce the incoming object. Initially, $\mathcal{N} = \emptyset$. Our algorithm does the following on receiving a new input object $\sigma$. If the existing piercing set pierces $\sigma$, do nothing. Otherwise, our online algorithm adds the center of $\sigma$ to $\mathcal{N}$.*

The analysis of the algorithm is similar in nature for fat objects, axis-aligned boxes and ellipsoids in $\mathbb{R}^d$. To bound the competitive ratio, we determine the number of points placed by algorithm against each point $p$ in an offline optimum. To compute the number of piercing points placed by our algorithm, we consider the region containing all objects that can be pierced by the point $p$. We have partitioned this region into $O(\log M)$ regions such that our algorithm places the same number of piercing points in each of these regions. Finally, we give an upper bound on the total number of points placed by our algorithm in each of these regions. The competitive ratio is $O(\log M)$ multiplied by this number. Proofs not included in the main body of this section are presented in Appendix B.

## 4.1 ONLINE PIERCING FOR AXIS-ALIGNED BOXES IN $\mathbb{R}^d$

In this section, we study the piercing set problem for $(\mathcal{X}, \mathcal{R})$, where the set $\mathcal{X}$ is the entire $\mathbb{R}^d$ and $\mathcal{R}$ is a family of axis-aligned arbitrary boxes from $[1, N]^d$ having side lengths in $[1, M]$. Note that boxes can have arbitrary aspect ratios, thus they are not necessarily fat objects. Hence, the result for piercing $\alpha$-fat objects (De et al. (2024b)) does not apply to boxes in $\mathbb{R}^d$. In fact, surprisingly, no online algorithm is known to date even for rectangles in $\mathbb{R}^2$. For a fixed $d \in \mathbb{Z}^+$, for piercing axis-aligned boxes in $\mathbb{R}^d$, we propose a simple deterministic algorithm ALGO-CENTER which obtains a competitive ratio $O(\log M))$ (Theorem 6). There exists a randomized lower bound for hypercubes in $\mathbb{R}^d$ of $\Omega(\log M)$ (De et al. (2024b)), which also holds for axis-aligned rectangles in $\mathbb{R}^d$. Thus, the competitive ratio obtained by our algorithm is asymptotically tight.

In this section, we first present the analysis of ALGO-CENTER for rectangles in $\mathbb{R}^2$. Later, we generalize it for higher dimensions (see 4.1.1). Throughout the section, all distances are $L_\infty$ distances, and all boxes are axis-aligned, unless stated otherwise.

**Theorem 5.** *For piercing axis-aligned rectangles in $\mathbb{R}^2$ having the length of each side in the range $[1, M]$, ALGO-CENTER has a competitive ratio of at most $O(\log M)$.*

*Proof.* Let $\mathcal{I}$ be the set of rectangles presented to the online algorithm. Let $\mathcal{N}$ and OPT denote the piercing set returned by the online algorithm ALGO-CENTER and an offline optimal for $\mathcal{I}$. Consider a point $p \in$ OPT and let $\mathcal{I}^p \subseteq \mathcal{I}$ be the collection of all the rectangles arrived so far and contain the point $p$. Let $\mathcal{N}^p \subseteq \mathcal{N}$ be the set of piercing points placed by ALGO-CENTER to pierce all the input rectangles in $\mathcal{I}^p$. Clearly, we have $\mathcal{N} = \bigcup_{p \in \text{OPT}} \mathcal{N}^p$. Consequently, the competitive ratio of our algorithm is upper bounded by $\max_{p \in \text{OPT}} |\mathcal{N}^p|$.

Now, consider any point $a \in \mathcal{N}^p$. Since $a$ is the center of a rectangle $\sigma \in \mathcal{I}^p$ that contains the point $p$ and has a side length of at most $M$, the distance between $a$ and $p$ is at most $\frac{M}{2}$. As a result, a square $S$ of side length $M$, centered at $p$, will contain all the points in $\mathcal{N}^p$. Next, partition the square $S$ into $(\lfloor \log M \rfloor + 1)$ smaller nested squares. For $i \in [(\lfloor \log M \rfloor + 1)]$, let $S_i$ be a square with sides of length $\frac{M}{2^{i-1}}$. Define the *annular region* $A_i = S_i \setminus S_{i+1}$, where $i \in [(\lfloor \log M \rfloor + 1)]$. Notice that the annular region $A_i$ contains all the rectangles of $\mathcal{I}_p$ whose length of both the sides are at least $\frac{M}{2^{i-1}}$. Let $\mathcal{N}_i^p = \mathcal{N}^p \cap A_i$ be the subset of $\mathcal{N}^p$ that is contained in the region $A_i$.

**Lemma 1** ($\star$). $|\mathcal{N}_i^p| \leq 12$.

Since $\bigcup \mathcal{N}_i^p = \mathcal{N}_p$ and due to Lemma 1 we have $\mathcal{N}_i^p \leq 12$, therefore $|\mathcal{N}_p| \leq 12 \times (\lfloor \log M \rfloor + 1) = O(\log M)$. Hence, the theorem follows. $\qquad\square$

#### 4.1.1 GENERALIZATION TO HIGHER DIMENSIONAL BOXES

Similar to $\mathbb{R}^2$, we have a hypercube $H$ of side length $M$ centered at $p \in \mathsf{OPT}$, containing all the centers of the objects in $\mathcal{I}_p$. We can partition the hypercube $H$ into $\lfloor \log M \rfloor + 1$ smaller hypercubes $S_i$. Specifically, the hypercube $S_i$ has all sides is of length $\frac{M}{2^{i-1}}$. Similar to the two dimensional case, here also we define the annular region $A_i = S_i \setminus S_{i+1}$, where $i \in [(\lfloor \log M \rfloor + 1)]$. Notice that the annular region $A_i$ contains all the boxes of $\mathcal{I}_p$ such that the length of all the sides are at least $\frac{M}{2^{i-1}}$. Let $\mathcal{N}_i^p = \mathcal{N}^p \cap A_i$ be the subset of $\mathcal{N}^p$ that is contained in the region $A_i$. For each $i \in [(\lfloor \log M \rfloor + 1)]$, we show that $|\mathcal{N}_i^p| \leq 2^d(2^d - 1) = O(4^d)$ (due to Lemma 2). Since $\cup \mathcal{N}_i^p = \mathcal{N}_p$. Thus, we have the following theorem.

**Theorem 6** (⋆). *For a fixed $d \in \mathbb{Z}^+$, for piercing arbitrary box in $\mathbb{R}^d$ having the length of each side in $[1, M]$,* ALGO-CENTER *has a competitive ratio of at most $O(\log M)$.*

### 4.2 ONLINE PIERCING FOR ELLIPSOID IN $\mathbb{R}^d$

In this section, we study the piercing set problem for a family of ellipsoids having length of axis-aligned semi-major and semi-minor axes in $[1, M]$, where $M > 1$. Similar to rectangles, ellipses can also have arbitrary aspect ratios and are not assumed to be fat. Surprisingly, no online algorithm is known to date, even for ellipses in $\mathbb{R}^2$. In this section, for a fixed $d \in \mathbb{Z}^+$, we show that for piercing ellipsoids in $\mathbb{R}^d$,. ALGO-CENTER achieves a competitive ratio of at most $O(\log M)$ (Theorem 8). The competitive ratio obtained by our algorithm is asymptotically tight, due to lower bound of De et al. (2024a) for ball is $\Omega(\log M)$.

Here, we first present the analysis of the algorithm for the case of ellipses in $\mathbb{R}^2$. Later, in Section 4.2.1, we generalize the analysis of the algorithm for the higher dimensional ellipsoids (see 4.2.1). The proof of the following theorem is similar to the proof of Theorem 6.

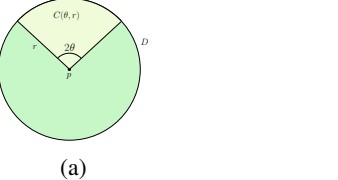 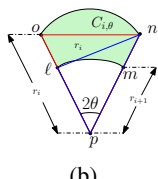

(a)  (b)

Figure 3: (a) Partitioning the disk $D$ of radius $r$ using circular sector $C(\theta, r)$; (b) Description of circular sector $C(\theta, r_i)$ and circular block $C_{i, \theta}$.

**Theorem 7** (⋆). *For piercing ellipses in $\mathbb{R}^2$ having length of axis aligned semi-major and semi-minor axis in the range $[1, M]$,* ALGO-CENTER *achieves a competitive ratio of at most $O(\log M)$.*

#### 4.2.1 GENERALIZATION TO HIGHER DIMENSIONAL ELLIPSOIDS

Similar to the two-dimensional case, we construct a $d$-dimensional ball $B$ of radius $M$ centered at $p \in \mathsf{OPT}$, containing all the centers of the $d$-dimensional ellipsoids in $\mathcal{I}_p$. We can partition the $d$-dimensional ball $B$ into $\lfloor \log M \rfloor + 1$ smaller concentric $d$-dimensional balls $B_i$. Specifically, the $d$-dimensional ball ball $B_i$ has radius $\frac{M}{2^{i-1}}$. Similar to the two dimensional case, here also we define the annular region $A_i = B_i \setminus B_{i+1}$, where $i \in [(\lfloor \log M \rfloor + 1)]$. Notice that the annular region $A_i$ contains all the $d$-dimensional ellipsoids of $\mathcal{I}_p$ such that the length of all the principal semi-axes is at least $\frac{M}{2^{i-1}}$. We prove that for each $i \in [(\lfloor \log M \rfloor + 1)]$, we have $|\mathcal{N}_i^p| \leq \left( \left( 1 + \frac{1}{\sin(\theta/2)} \right)^d - 1 \right)$, where $\theta = \frac{1}{2} \cos^{-1} \left( \frac{1}{2} + \frac{1}{1+\sqrt{1+4\alpha^2}} \right)$ and $x = \frac{\sqrt{5}-1}{2}$. Since $\cup \mathcal{N}_i^p = \mathcal{N}_p$. Thus, similar to Theorem 7, we have the following theorem.

**Theorem 8** (⋆). *For a fixed $d \in \mathbb{Z}^+$, for piercing $d$-dimensional ellipsoids having the length of all the axis-aligned principal semi-axes in $[1, M]$,* ALGO-CENTER *has a competitive ratio of at most $O(\log M)$.*

### 4.3 ONLINE PIERCING FOR $\alpha$-FAT OBJECTS IN $\mathbb{R}^d$

In this section, we focus on piercing $\alpha$-fat objects in $\mathbb{R}^d$. Currently, the best known bound on competitive ratio is $O\left( (\frac{2}{\alpha} + 2)^d \log M \right)$ (De et al. (2024a)). We improve this result for $(\alpha \in$

$\frac{1}{2}, 1]$ by introducing a simple deterministic algorithm with a slightly better competitive ratio of $O\left(\left(\frac{2}{\alpha} + \frac{7}{8}\right)^d \log M\right)$. This resolves an open problem posed by De et al. (2024a), which seeks to narrow the gap between the lower and upper bounds for piercing $\alpha$-fat objects in higher dimensions. We consider all the distances in this section to be under $L_\infty$-norm, unless stated otherwise.

Before describing the algorithm, we first present some essential ingredients that will be used for describing the algorithm. For any $j \in 2[\lfloor \log M \rfloor] \cup \{0\}$, let $\ell_j = 2.2^{\frac{j}{2}+1}$ and $u_j = 3.2^{\frac{j}{2}+1}$ if $j$ is even, and $\ell_j = 3.2^{\frac{j-1}{2}+1}$ and $u_j = 4.2^{\frac{j-1}{2}+1}$ if $j$ is odd.

**Layer of the objects.** We partition the set of all similarly-sized fat objects into $[2\lfloor \log M \rfloor + 1] \cup \{0\}$ layers. When $j$ is even (respectively, odd), the layer $L_j$ contains the fat objects having widths in $[\ell_j, u_j)$.

**Lattice.** Let $\Pi_d^j = \{\alpha_1 \ell_j \mathbf{e}_1 + \alpha_2 \ell_j \mathbf{e}_2 + \ldots + \alpha_d \ell_j \mathbf{e}_d \mid (\alpha_1, \alpha_2, \ldots, \alpha_d) \in \mathbb{Z}^d\}$ be a $d$-dimensional lattice spanned by the standard unit vectors. To visualize $\Pi_1^j, \Pi_2^j$ and $\Pi_3^j$, we refer to Figure 4.

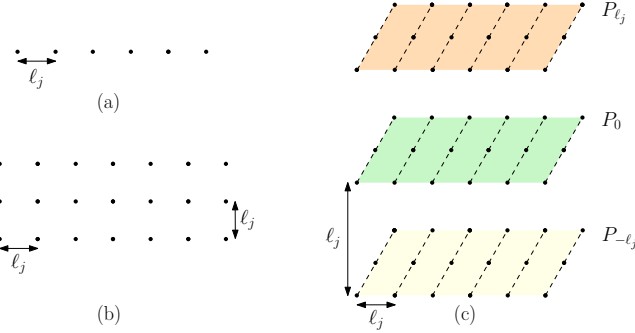

Figure 4: The points of $\Pi_d^j$ are drawn (a) for $d = 1$, (b) for $d = 2$, (c) for $d = 3$. In (c), the projections of planes $P_{\ell_j}$, $P_0$ and $P_{-\ell_j}$ over a rectangular region is depicted. Here, for any $k' \in \mathbb{R}$, $P_{k'} = \{y \in \mathbb{R}^d \mid y(x_d) = k'\}$ is a hyper-plane.

Now, we present a simple deterministic online algorithm for piercing fat objects in $\mathbb{R}^d$.

**Online algorithm** ALGO-FAT. Let $\mathcal{N}$ be the piercing set maintained by our algorithm to pierce the incoming fat objects. Initially, $\mathcal{N} = \emptyset$. On receiving a new input object $\sigma$ with width $s$, we do the following. If it is already hit by $\mathcal{N}$, then ignore $\sigma$. Otherwise, first determine the layer $L_j$ in which $\sigma$ belongs, where $j = \log_{\frac{3}{2}} s$. Then, our algorithm choose the *closest point* $r$ from $\Pi_d^j \cap \sigma$, and adds $r$ to $\mathcal{H}$.

For the correctness, efficient implementation and analysis of the online algorithm, see Appendix B.3.

**Theorem 9** ($\star$). *For piercing similarly-sized fat objects with widths in $[1, M)$,* ALGO-FAT *has a competitive ratio of at most $O(\lfloor \frac{2}{\alpha} + \frac{7}{8} \rfloor^d \log M)$.*

## 5 CONCLUSION

We studied the *online $\varepsilon$-net* and *online piercing set* problems for a wide range of geometric objects. For the *online $\varepsilon$-net*, we have obtained asymptotically tight bounds for the competitive ratios for some of these objects. Two future directions particularly arise from our work. What happens to other geometric objects? We believe that some techniques used in this work could be extended to other related geometric objects of constant description complexity in $\mathbb{R}^d$, for $d \leq 3$. Obtaining tight bounds for all objects of bounded VC-dimension is an interesting open problem. Moreover, to ensure the cardinality of an optimal sample size, we used the value of $\varepsilon$ within a certain regime. Designing online algorithms for any $\varepsilon > 0$ is an interesting open problem. For *online piercing set*, we have established asymptotically tight bounds on the competitive ratios for piercing hyper-rectangles and ellipsoids in $\mathbb{R}^d$, for any $d \in \mathbb{N}$. A challenging open question remains whether it is possible to remove the dependence on the dimension from the competitive ratios bound for classes of objects of bounded VC-dimension.

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

# A    MISSING PROOFS OF SECTION 3

## A.1    MISSING PROOFS OF SECTION 3.1

**Theorem 2.** *For online $\varepsilon$-net of arbitrary intervals, there exists a deterministic online algorithm with a competitive ratio of $2\left(\log\left(\frac{1}{\varepsilon}\right)+1\right)$, for any $\varepsilon \in (0,1]$. This result is tight: the competitive ratio of any deterministic online algorithm for this problem is at least $\log\frac{1}{\varepsilon}+1$.*

*Proof.* **Upper Bound.** Let $\mathcal{I}$ be a collection of intervals presented to the online algorithm. Let $\mathcal{S} \subseteq \mathcal{I}$ be the collection of intervals $[a,b]$ such that $|[a,b] \cap \mathcal{X}| \geq \varepsilon|\mathcal{X}|$. For each $i \in \left[\lceil \log_2 \frac{1}{\varepsilon}\rceil\right] \cup \{0\}$, let $\mathcal{S}_i$ be the collection of intervals $[x,y]$ in $\mathcal{S}$ such that $|[x,y] \cap \mathcal{X}| \in \left[2^i\varepsilon|\mathcal{X}|, 2^{i+1}\varepsilon|\mathcal{X}|\right)$. Let $\mathcal{N}$ and $\mathsf{OPT}$ denote the sub-collection of $\varepsilon$-nets returned by the online algorithm and an optimal offline algorithm, respectively, for $\mathcal{S}$. Let $p$ be a point in $\mathsf{OPT}$. Let $\mathcal{S}(p) \subseteq \mathcal{S}$ be the set of intervals that contains the point $p$. For each $i \in \left[\lceil \log_2 \frac{1}{\varepsilon}\rceil\right] \cup \{0\}$, let $\mathcal{S}_i(p) = \mathcal{S}(p) \cap \mathcal{S}_i$. Let $\mathcal{N}_i(p) \subseteq \mathcal{N}$ be the set of points that are placed by the online algorithm to hit an input interval in $\mathcal{S}_i(p)$ which is not hit. We claim that our online algorithm places at most 2 points for all intervals in $\mathcal{S}_i(p)$. Without loss of generality, let us assume that $\sigma \in \mathcal{S}_i(p)$ is the first input interval that is not hit upon its arrival. In order to hit $\sigma$, our online algorithm adds the points indexed $\left\lfloor \frac{|\sigma \cap \mathcal{X}|}{2}\right\rfloor$ and $\left\lceil \frac{|\sigma \cap \mathcal{X}|}{2}\right\rceil$ to $\mathcal{N}_i(p)$. Notice that we can partition $\sigma$ into two disjoint sub-intervals $P_\sigma^\ell$ and $P_\sigma^r$ such that they contain $\left\lfloor \frac{|\sigma \cap \mathcal{X}|}{2}\right\rfloor$ and $\left\lceil \frac{|\sigma \cap \mathcal{X}|}{2}\right\rceil$ points, respectively. Let $P_\sigma^t$ contains the point $p$, where $t \in \{\ell, r\}$. Let $\sigma'(\neq \sigma) \in \mathcal{S}_i(p)$ be any interval. Observe that, $|P_\sigma^t| < \left\lfloor \frac{|\sigma \cap \mathcal{X}|}{2}\right\rfloor < 2^i\varepsilon|\mathcal{X}|$. Also, by definition, $\sigma' \in \mathcal{S}_i(p)$ contains at least $2^i\varepsilon|\mathcal{X}|$ points, and $\sigma' \cap P_\sigma^t \neq \emptyset$. As a result, $\sigma'$ hit by either the point of $\sigma$ indexed $\left\lfloor \frac{|\sigma \cap \mathcal{X}|}{2}\right\rfloor$ or $\left\lceil \frac{|\sigma \cap \mathcal{X}|}{2}\right\rceil$. Therefore, our algorithm does not add any point to $\mathcal{N}_j(p)$ for $\sigma'$. Thus, $\mathcal{N}(p)$ contains at most $2\left(\lceil \log_2 \frac{1}{\varepsilon}\rceil + 1\right)$ points.

**Lower Bound.** To prove the lower bound, we can think it as a game between the adaptive adversary and online algorithm. Let $\sigma_1$ be the first interval containing all the $n$ points presented by the adversary to the online algorithm. Let $p_1$ be a point placed by the online algorithm to pierce the interval $\sigma_1$. The point $p_1$ partitions the interval $\sigma_1$ into two parts, of which, let $\sigma_1^L$ be a part containing at least $\frac{n}{2}$ points that does not contain the point $p_1$. Now, the adversary can present an interval $\sigma_2 \subseteq \sigma_1^L$ containing exactly $\frac{n}{2}$ points. For the new interval $\sigma_2$, any online algorithm needs a new piercing point $p_2$. Now again, one can define a partition $\sigma_2^L$ of $\sigma_2$ depending on the position of the point $p_2$ such that $\sigma_2^L$ contains at least $\frac{n}{4}$ points and does not contain $p_2$. The adversary will present an interval $\sigma_3 \subseteq \sigma_2^L$ containing exactly $\frac{n}{4}$ points. In this way, the adversary can adaptively construct $\log\frac{1}{\varepsilon} + 1$ intervals (last interval containing $\varepsilon n$ points) for which any online algorithm needs $\log\frac{1}{\varepsilon} + 1$ distinct points to pierce, while an offline optimum needs only one point. Hence, the lower bound of the competitive ratio is $\log\frac{1}{\varepsilon} + 1$.

Hence, we conclude the proof of the theorem. □

## A.2    MISSING PROOFS OF SECTION 3.2

**Theorem 4.** *For the online $\varepsilon$-net problem with arbitrary axis-aligned boxes in $\mathbb{R}^3$, there exists an algorithm with an expected competitive ratio of at most $O\left(\log^3\left(\frac{1}{\varepsilon}\right)\right)$. Here, $\varepsilon \in \left(\frac{1}{C}, 1\right]$, where $C$ is a sufficiently large constant.*

Before presenting the proof of the theorem, we first define some important ingredients that will play an essential role in proving the above-mentioned theorem.

We begin by selecting a random sample $\mathcal{P} \subseteq \mathcal{X}$ in $\mathbb{R}^3$ of size $O\left(\varepsilon \log \log \frac{1}{\varepsilon}\right)$, similar to the case in $\mathbb{R}^2$. Then, we construct a three-level range tree $\mathcal{T}$ over the points of $\mathcal{X}$. This can be obtained by following standard methods from Computational Geometry (see Berg et al. (2008)). The construction of the range tree proceeds as follows: in the primary tree, the points are sorted by their $x$-coordinates,

in the secondary tree by their $y$-coordinates, and in the tertiary tree by their $z$-coordinates. Each node $u$ in the primary tree $\mathcal{T}$ is associated with a subset $\mathcal{R}_u \subseteq \mathcal{X}$, and it a secondary tree $\mathcal{T}_u$ on this subset. Each node $v$ in $\mathcal{T}_u$ is similarly associated with a subset $\mathcal{R}_{u,v} \subseteq \mathcal{R}_u$ and a tertiary tree $\mathcal{T}_{u,v}$ on this subset. Finally, each node $w$ in a tertiary tree $\mathcal{T}_{u,v}$ corresponds to a subset $\mathcal{R}_{u,v,w} \subseteq \mathcal{R}_{u,v}$.

For each internal node $u$ in the primary tree, we associate a plane $h_u$ orthogonal to the $x$-axis that splits the points into subsets and stored at its children (analogous to the line $l_u$ for each node $u$). Similarly, for each node $v$ in $\mathcal{T}_u$ and each node $w$ in $\mathcal{T}_{u,v}$, we associate planes $h_{u,v}$ and $h_{u,v,w}$ orthogonal to the $y$-axis and $z$-axis, respectively. Trees of each level has a depth of at most $O(\log \frac{1}{\varepsilon})$, and the total number of nodes in the range tree $\mathcal{T}$ is $O\left(\frac{1}{\varepsilon} \log^2 \frac{1}{\varepsilon}\right)$.

These orthogonal planes define octants $s_{u,v,w}$ for each node $w$ of the tertiary tree, analogous to the strips in the $\mathbb{R}^2$. For each octant, we construct a set $\mathcal{M}_{u,v,w}$ of maximal $\mathcal{P}$-unhit boxes. Each box $M$ requires at most three points from $\mathcal{P}_{u,v,w}$ to define its boundaries on each distinct facets. Similar to the case in $\mathbb{R}^2$, the number of maximal boxes $|\mathcal{M}_{u,v,w}|$ is at most $|\mathcal{P}_{u,v,w}| + 1 = O(|\mathcal{P}|)$. Therefore, the total number of maximal boxes across all octants is not more than $O\left(\frac{1}{\varepsilon} \log^2 \frac{1}{\varepsilon} \cdot |\mathcal{P}|\right)$.

To handle an input box $\sigma$ that is $\varepsilon$-heavy but not hit by $\mathcal{P}$, we follow a similar procedure like the $\mathbb{R}^2$: Identify the lowest-level plane intersecting $\sigma$, extend $\sigma$ to the boundary of the octant or until it intersects a point from $\mathcal{P}_{u,v,w}$, and form a box that belongs to the set $\mathcal{M}_{u,v,w}$. Finally, we define weights $w_M$ for each $M \in \mathcal{M}_{u,v,w}$ and construct safety nets $N_M$ as in the planar case. The final $\varepsilon$-net in $\mathbb{R}^3$ is the union of $\mathcal{P}$ with all the safety nets $N_M$. Using the similar reasoning described in Section 3.2, it is possible to show that the constructed set $N$ is indeed an $\varepsilon$-net, where $\varepsilon \in [1/C, 1)$ for any sufficiently large constant $C > 1$. Now, the proof of Theorem 4 is as follows.

*Proof.* The expected size of $N$ is given as,
$\mathbb{E}[|N|] = \mathbb{E}\left[|\mathcal{P}'| + \sum_{u \in \mathcal{T}} \sum_{v \in \mathcal{T}_v} \sum_{w \in \mathcal{T}_{u,v}} \sum_{M \in \mathcal{M}_{u,v,w}} w_M \log w_M\right]$, which can be bounded as $\mathbb{E}[|\mathcal{P}'|] + O\left(|\mathcal{P}| \cdot \frac{1}{\varepsilon} \log^2 \frac{1}{\varepsilon}\right) w_M \log w_M$.

Finally, applying the lower bound results from Pach & Tardos (2011), one can establish that the competitive ratio for the $\mathbb{R}^3$ case is:

$$\frac{\mathbb{E}[|N|]}{|\mathcal{OPT}|} \leq \frac{O\left(\log^2 \frac{1}{\varepsilon} \cdot \log \log \frac{1}{\varepsilon} \times O\left(\frac{1}{\varepsilon} \log \frac{1}{\varepsilon}\right)\right)}{O\left(\frac{1}{\varepsilon} \log \log \frac{1}{\varepsilon}\right)} = O\left(\log^3 \frac{1}{\varepsilon}\right)$$

This shows that the competitive ratio is $O\left(\log^3 \frac{1}{\varepsilon}\right)$. $\qquad\square$

## B    MISSING PROOFS OF SECTION 4

### B.1    MISSING PROOFS OF SECTION 4.1

**Lemma 1.** $|\mathcal{N}_i^p| \leq 12$.

*Proof.* Since $A_i = S_i \setminus S_{i+1}$, the distance (under $L_\infty$ norm) from the center $p$ to the boundary of $S_i$ and $S_{i+1}$ is $\frac{M}{2^{i-1}}$ and $\frac{M}{2^i}$, respectively. Thus, the annular region $A_i$ can contain squares of side length $\frac{M}{2^{i+1}}$.

**Claim 1.** *The annular region $A_i$ is the union of at most 12 disjoint squares, each having side length* $\frac{M}{2^{i+1}}$.

*Proof.* To calculate the number of such squares $S$ of side length $\frac{M}{2^{i+1}}$ in the annular region $A_i$, we need to find the ratio of area of $A_i$ with respect to the area of square $S$. The area of $A_i$ is equals to the area of $S_i$ minus the area of $S_{i+1}$. Thus, the area of $A_i$ is $\frac{3M^2}{4}$. The area of $S$ is $\frac{M^2}{16}$. Thus the ratio will be 12. Hence, the claim follows. $\qquad\square$

To complete the proof, next, we argue that our online algorithm places at the most one piercing point in each of these squares to pierce the objects in $\mathcal{I}^p$. Let $S$ be any such square of side length $\frac{M}{(2)^{i+1}}$,

and let $q_1 \in S$ be a piercing point placed by our online algorithm. For a contradiction, let us assume that our online algorithm places another piercing point $q_2 \in H$, where $q_2$ is the center of an object $\sigma \in \mathcal{I}_p$. Since $\sigma$ contains both the points $p$ and $q_2$, the distance (under $L_\infty$ norm) between them is at least $\frac{M}{2^{i_1}}$. Note that the distance (under $L_\infty$ norm) between any two points in $S$ is at most $\frac{M}{2^{i+1}}$, as a result, $\sigma$ is already pierced by $q_1$, since $\sigma$ is a rectangle of side length at least $\frac{M}{2^i}$. This contradicts our algorithm. Thus, the region $H$ contains at most one piercing point of $\mathcal{N}_i^p$. Hence, the lemma follows. $\qquad \square$

**Lemma 2.** $|\mathcal{N}_i^p| \leq 2^d(2^d - 1) = O(4^d)$.

*Proof.* Since $A_i = S_i \setminus S_{i+1}$, the distance (under $L_\infty$ norm) from the center $p$ to the boundary of $S_i$ and $S_{i+1}$ is $\frac{M}{2^{i-1}}$ and $\frac{M}{2^i}$, respectively. Thus, the annular region $A_i$ can contain squares of side length $\frac{M}{2^{i+1}}$.

**Claim 2.** *The annular region $A_i$ is the union of at most $12$ disjoint squares, each having side length $\frac{M}{2^{i+1}}$.*

*Proof.* To calculate the number of such hypercubes $H$ of side length $\frac{M}{2^{i+1}}$ in the annular region $A_i$, we need to find the ratio of the volume of $A_i$ with respect to the volume of hypercube $S$. The area of $A_i$ is equals to the volume of $S_i$ minus the volume of $S_{i+1}$. Thus, the area of $A_i$ is $\frac{(2^d-1)M^d}{2^d}$. The area of $S$ is $\frac{M^2}{4^d}$. Thus the ratio will be $2^d(2^d - 1)$. Hence, the claim follows. $\qquad \square$

To complete the proof, next, we argue that our online algorithm places at the most one piercing point in each of these squares to pierce the objects in $\mathcal{I}^p$. Let $S$ be any such square of side length $\frac{M}{(2)^{i+1}}$, and let $q_1 \in S$ be a piercing point placed by our online algorithm. For a contradiction, let us assume that our online algorithm places another piercing point $q_2 \in H$, where $q_2$ is the center of an object $\sigma \in \mathcal{I}_p$. Since $\sigma$ contains both the points $p$ and $q_2$, the distance (under $L_\infty$ norm) between them is at least $\frac{M}{2^{i_1}}$. Note that the distance (under $L_\infty$ norm) between any two points in $S$ is at most $\frac{M}{2^{i+1}}$, as a result, $\sigma$ is already pierced by $q_1$, since $\sigma$ is a rectangle of side length at least $\frac{M}{2^i}$. This contradicts our algorithm. Thus, the region $H$ contains at most one piercing point of $\mathcal{N}_i^p$. Hence, the lemma follows. $\qquad \square$

### B.2 MISSING PROOFS OF SECTION 4.2

**Theorem 7.** *For piercing ellipses in $\mathbb{R}^2$ having length of axis aligned semi-major and semi-minor axis in the range $[1, M]$, ALGO-CENTER achieves a competitive ratio of at most $O(\log M)$.*

*Proof.* Let $\mathcal{I}$ be the set of input ellipses in $\mathbb{R}^2$ presented to the algorithm. Let $\mathcal{N}$ and OPT be two piercing sets for $\mathcal{I}$ returned by ALGO-CENTER and the offline optimal, respectively. Let $p$ be any piercing point of the offline optimal OPT. Let $\mathcal{I}_p \subseteq \mathcal{I}$ be the set of input ellipses pierced by the point $p$. Let $\mathcal{N}_p$ be the set of piercing points placed by our algorithm to pierce all the ellipses in $\mathcal{I}_p$. To prove the theorem, we will give an upper bound of $|\mathcal{N}_p|$.

Let us consider any point $a \in \mathcal{N}_p$. Since $a$ is the center of an ellipse $\sigma \in \mathcal{I}_p$ (containing the point $p$) having length of semi-minor and semi-major axes at most $M$, the distance between $a$ and $p$ is at most $\frac{M}{2}$. Therefore, a disk $D$ of radius $M$, centered at $p$, contains all the points in $\mathcal{N}_p$. Let $x = \frac{\sqrt{5}-1}{2}$ be a positive constant. Let $D_i$ be a disk centered at $p$ having radius $r_i = \frac{M}{(1+x)^{i-1}}$, where $i \in [(\lfloor \log M \rfloor + 1)]$. Note that $D_1, D_2, \ldots, D_m$ are concentric disks, centered at $p$. Let $\theta = \frac{1}{2}\cos^{-1}\left(\frac{1}{2} + \frac{1}{1+\sqrt{5}}\right)$ be a constant angle in $(0, \frac{\pi}{10}]$. Similar to the case of rectangles, now we define the annular region $A_i = D_i \setminus D_{i+1}$. Let $C(\theta, r_i)$ be a *circular sector* obtained by taking the portion of the disk $D_i$ by a conical boundary with the apex at the center $p$ of the disk and $\theta$ as the half of the cone angle (for an illustration see Figure 3a). For any $i \in [\lfloor \log M \rfloor + 1]$, let us define the $i$th *circular block* $C_{i,\theta} = C(\theta, r_i) \setminus C(\theta, r_{i+1})$ (for an illustration see Figure 3b). Notice that all the $i$th circular blocks contain all the ellipses of $\mathcal{I}_p$ having length of both the semi-major and semi-minor axes at least $r_i$.

Similar to Lemma 1, we have the following lemma.

**Lemma 3.** $|\mathcal{N}_i^p| \leq \lceil \frac{\pi}{\theta} \rceil$.

*Proof.* Notice that the total angle of any disk $\mathcal{D}$ centered at $p$ is $2\pi$. If any cone having apex at $p$ and angle $2\theta$, then at most $\lceil \frac{\pi}{\theta} \rceil$ cones will cover the entire $\mathcal{D}$. Thus, it is easy to observe that $\left\lceil \frac{2\pi}{2\theta} \right\rceil$ circular blocks will entirely cover $A_i$, there are at most $\frac{\pi}{\theta}$ circular blocks $C_{i,\theta}$ in $A_i$. Now, we will show that in each circular block our algorithm places only one point. Let $q_1$ be the first piercing point placed by ALGO-CENTER in $C_{i,\theta}$. For a contradiction, let us assume that ALGO-CENTER places another piercing point $q_2 \in C_{i,\theta}$, where $q_2$ is center of some ellipse $\sigma \in \mathcal{I}_p$. Since $\sigma$ contains points $p$ and $q_2$, and the distance between them is at least $r_i$. Note that the maximum distance between any two points in $C_{i,\theta}$ is at most $\max\{\overline{ln}, \overline{on}\}$. It is easy to observe that $\max\{\overline{ln}, \overline{on}\}$ is at most $r_i$. As a result, $\sigma$ is already pierced by $q_1$. This contradicts that our algorithm places two piercing points in $C_{i,\theta}$. Hence, ALGO-CENTER places at most one piercing point in the circular block $C_{i,\theta}$ to pierce an ellipse in $\mathcal{I}_p$. Hence, the lemma follows. □

Since $\cup \mathcal{N}_i^p = \mathcal{N}_p$ and due to Lemma 3 we have $|\mathcal{N}_i^p| \leq \frac{\pi}{\theta}$, therefore $|\mathcal{N}_p| \leq \lceil \frac{\pi}{\theta} \rceil \times (\lfloor \log M \rfloor + 1) = O(\log M)$. Hence, the theorem follows. □

**Theorem 8.** *For a fixed $d \in \mathbb{Z}^+$, for piercing $d$-dimensional ellipsoids having the length of all the axis-aligned principal semi-axes in $[1, M)$, ALGO-CENTER has a competitive ratio of at most $O(\log M)$.*

*Proof.* Let $\mathcal{I}$ be the set of input ellipsoids in $\mathbb{R}^d$ presented to the algorithm. Let $\mathcal{N}$ and OPT be two piercing sets for $\mathcal{I}$ returned by ALGO-CENTER and the offline optimal, respectively. Let $p$ be any piercing point of the offline optimal OPT. Let $\mathcal{I}_p \subseteq \mathcal{I}$ be the set of input ellipsoids pierced by the point $p$. Let $\mathcal{N}_p$ be the set of piercing points placed by our algorithm to pierce all the ellipsoids in $\mathcal{I}_p$. To prove the theorem, we will give an upper bound of $|\mathcal{N}_p|$.

Let us consider any point $a \in \mathcal{N}_p$. Since $a$ is the center of an ellipsoids $\sigma \in \mathcal{I}_p$ (containing the point $p$) having length of principal semi-axes is at most $M$, the distance between $a$ and $p$ is at most $\frac{M}{2}$. Therefore, a ball $B$ of radius $M$, centered at $p$, contains all the points in $\mathcal{N}_p$. Let $x = \frac{\sqrt{5}-1}{2}$ be a positive constant. Let $D_i$ be a disk centered at $p$ having radius $r_i = \frac{M}{(1+x)^{i-1}}$, where $i \in [(\lfloor \log M \rfloor + 1)]$. Note that $D_1, D_2, \ldots, D_m$ are concentric balls, centered at $p$. Let $\theta = \frac{1}{2} \cos^{-1} \left( \frac{1}{2} + \frac{1}{1+\sqrt{5}} \right)$ be a constant angle in $(0, \frac{\pi}{10}]$. Similar to the case of rectangles, now we define the annular region $A_i = D_i \setminus D_{i+1}$. Let $H(\theta, r_i)$ be a *hyper-spherical sector* obtained by taking the portion of the ball $B_i$ by a conical boundary with the apex at the center $p$ of the ball and $\theta$ as the half of the cone angle. For any $i \in [\lfloor \log M \rfloor + 1]$, let us define the $i$th *hyper-spherical block* $H_{i,\theta} = H(\theta, r_i) \setminus H(\theta, r_{i+1})$. Notice that all the $i$th hyper-spherical blocks contain all the ellipsoids of $\mathcal{I}_p$ having length of all the principal semi-axes is at least $r_i$.

Since $\cup \mathcal{N}_i^p = \mathcal{N}_p$ and due to Lemma 3 we have $|\mathcal{N}_i^p| \leq \left( \left(1 + \frac{1}{\sin(\theta/2)}\right)^d - 1 \right)$, therefore $|\mathcal{N}_p| \leq \left( \left(1 + \frac{1}{\sin(\theta/2)}\right)^d - 1 \right) \times (\lfloor \log M \rfloor + 1) = O(\log M)$. Hence, the theorem follows. □

Similar to Lemma 1, we have the following lemma.

**Lemma 4.** $|\mathcal{N}_i^p| \leq \left( \left(1 + \frac{1}{\sin(\theta/2)}\right)^d - 1 \right)$, *where* $\theta = \frac{1}{2} \cos^{-1} \left( \frac{1}{2} + \frac{1}{1+\sqrt{1+4\alpha^2}} \right)$ *and* $x = \frac{\sqrt{5}-1}{2}$.

*Proof.* Due to (Devroye et al., 1996, Lemma 5.3), for any fixed $\theta \in (0, \pi/2)$, we need at most $\left( \left(1 + \frac{1}{\sin(\theta/2)}\right)^d - 1 \right)$ hyper-cones with angle $2\theta$ completely cover $\mathbb{R}^d$. As a result, for any fixed $\theta \in (0, \pi/2)$, we need at most $\left( \left(1 + \frac{1}{\sin(\theta/2)}\right)^d - 1 \right)$ hyper-spherical blocks $H(i, \theta)$ to completely

cover the annular region $A_i$. Now, we will show that in each hyper-spherical block our algorithm places only one point. Let $q_1$ be the first piercing point placed by ALGO-CENTER in $H_{i,\theta}$. For a contradiction, let us assume that ALGO-CENTER places another piercing point $q_2 \in H_{i,\theta}$, where $q_2$ is center of some ellipsoid $\sigma \in \mathcal{I}_p$. Since $\sigma$ contains points $p$ and $q_2$, and the distance between them is at least $r_i$. Due to Claim 3 the maximum distance between any two points in $H_{i,\theta}$ is at most $r_i$. As a result, $\sigma$ is already pierced by $q_1$. This contradicts that our algorithm places two piercing points in $H_{i,\theta}$. Hence, ALGO-CENTER places at most one piercing point in the hyper-spherical block $H_{i,\theta}$ to pierce an ellipse in $\mathcal{I}_p$. Hence, the lemma follows. □

**Claim 3.** *The distance between any two points in $H_{i,\theta}$ is at most $r_i$.*

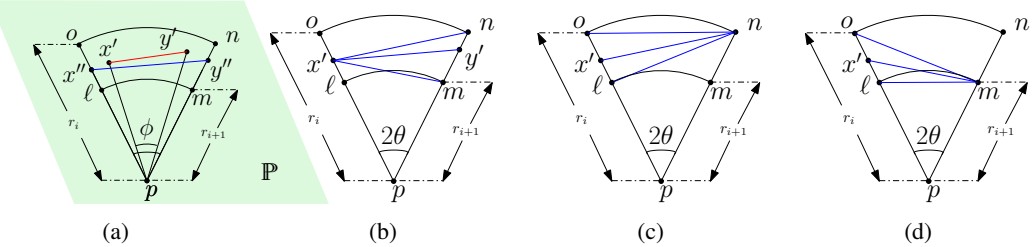

Figure 5: (a) Description of the plane $\mathbb{P}$. (b) Illustration of triangles $\triangle my'x'$ and $\triangle ny'x'$. (c) Illustration of triangles $\triangle oox'n$ and $\triangle \ell x'n$ (d) Illustration of triangles $\triangle ox'm$ and $\triangle x'\ell m$, in $T_{i,\theta}$.

*Proof.* Observe Figure 5, where a detail of the projection hyper-spherical block is depicted. Note that the maximum distance between any two points in $T_{i,\theta}$ is at most $\max\{\overline{ln}, \overline{on}\}$. First, consider the triangle $\triangle \ell pn$ (see Figure 5). By the cosine rule of the triangle, we have:

$$
\begin{aligned}
\overline{ln}^2 &= \overline{p\ell}^2 + \overline{pn}^2 - 2\overline{p\ell}\,\overline{pn}\cos(2\theta) \\
&= \left(\frac{M}{(1+x)^{i-1}}\right)^2 + \left(\frac{M}{(1+x)^{i-2}}\right)^2 - 2\left(\frac{M}{(1+x))^{i-1}}\right)\left(\frac{M}{(1+x))^{i-2}}\right)\cos(2\theta) \\
&= \left(\frac{M}{(1+x)^{i-2}}\right)^2\left(\left(\frac{1}{(1+x)}\right)^2 + 1 - 2\left(\frac{1}{(1+x)}\cos(2\theta)\right)\right) \\
&= \left(\frac{M}{(1+x)^{i-1}}\right)^2\left(1 + (1+x)^2 - 2(1+x)\cos(2\theta)\right)
\end{aligned}
$$

Since $\theta = \frac{1}{2}\cos^{-1}\left(\frac{1}{2} + \frac{1}{1+\sqrt{5}}\right)$ and $x = \frac{\sqrt{5}-1}{2}$, $\cos(2\theta) = \frac{(x+2)}{2(x+1)}$ and $x^2 + x = 1$. Now substituting these values in the above equation, we get

$$
\overline{ln}^2 = r_i^2\left(1 + (1+x)^2 - (2+x)\right) = r_i^2\left(1 + 1 + x^2 + 2x - 2 - 2x\right) = r_i^2\left(x^2 + x\right) = (r_i)^2.
$$

Now, consider the triangle $\triangle opn$ (see Figure 5). Here we have:

$$
\begin{aligned}
\overline{on}^2 &= 2\left(\frac{M}{\alpha(1+x)^{i-2}}\right)^2 - 2\left(\frac{M}{\alpha(1+x)^{i-2}}\right)^2\cos(2\theta) = 2\left(\frac{M}{\alpha(1+x)^{i-2}}\right)^2(1 - \cos(2\theta)) \\
&= 2\left(\frac{M}{\alpha(1+x)^{i-1}}\right)^2(1+x)^2(1 - \cos(2\theta)) = 2r_i^2(1+x)^2(1 - \cos(2\theta)).
\end{aligned}
$$

Now substituting the values of $\cos(2\theta) = \frac{(x+2)}{2(x+1)}$ and $x^2 + x = 1$ in the above equation, we get

$$
\begin{aligned}
\overline{on}^2 &= 2r_i^2(1+x)^2\left(1 - \frac{(x+2)}{2(x+1)}\right) \\
&= 2r_i^2(1+x)^2\left(\frac{2(x+1) - (x+2)}{2(x+1)}\right) \\
&= r_i^2(1+x)x = (r_i)^2.
\end{aligned}
$$

Note that $\overline{ln} = \overline{on} = r_i$. Thus $r_i$ is the maximum distance between any two points in the region $H_{i,\theta}$. □

## B.3 Missing proofs of Section 4.3

**Efficient implementation of the algorithm.** For the efficient implementation of the algorithm, given a fat object $\sigma \in \mathbb{R}^d$ centered at $q$ with width $s$, it is crucial to determine the layer $L_j$ to which the fat object belongs. This can be done in $O(1)$ time, since $j = \log_{\frac{3}{2}} s$. Next, identifying the closest point from $\Pi_d^j$ to the fat object's centre $c$ is important, and according to the following lemma, it can be done in $O(d)$ time.

**Lemma 5.** *For any point $q$ in $\mathbb{R}^d$, there exists a point $r$ in $\Pi_d^j$ such that $d_\infty(q, r) \leq \frac{\ell_j}{2}$. Given the center, $q$ of the fat object, the closest point $r \in \Pi_d^j$ can be found in $O(d)$ time.*

*Proof.* Notice that for any point $r \in \Pi_d^j$, each coordinate of $r$ is an integral multiple of $\ell_j$. For any point $q \in \mathbb{R}^d$, for each $j \in [d]$, the $j$th coordinate of the point $q$ can be uniquely written as $q(x_j) = z_j + y_j$, where $z_j \in \ell_j \mathbb{Z}$ and $y_j \in [0, \ell_j)$. Here, by $\beta \mathbb{Z}$ we mean the set $\left\{ \beta z \mid z \in \mathbb{Z} \right\}$. Now, we define the best point $r$ of $\Pi_d^j$ depending on the coordinates of $q$. For each $j \in [d]$, we set the $j$th coordinate of $r$ as follows.

$$r(x_j) = \begin{cases} z_j, & \text{if } y_j \in [0, \frac{\ell_j}{2}) \\ z_j + \ell_j, & \text{if } y_j \in [\frac{\ell_j}{2}, \ell_j). \end{cases}$$

As per the construction of the point $r$, we have $|r(x_j) - q(x_j)| \leq \frac{\ell_j}{2}$ for each $j \in [d]$. As a result, $d_\infty(r, q) = \max_{j \in [d]} |r(x_i) - q(x_i)| \leq \frac{\ell_j}{2}$. □

**Correctness of the algorithm.** Due to Lemma 5, there exists a point $r \in \Pi_d^j$ such that $d_\infty(c, r) \leq \ell_j$. Recall that any fat object in $L_j$ has width at least $\ell_j$, it contains a hypercube with side length at least $\ell_j$. Thus, $\sigma \in L_j$ must contain at least $r$. Hence, the above-mentioned online algorithm is a feasible algorithm.

**Analysis of the algorithm.** Let $\mathcal{I}$ be a set of input fat objects presented to the algorithm. For each $j \in [2\lfloor \log M \rfloor + 1] \cup \{0\}$, let $\mathcal{I}_j$ be the collection of all fat objects in $\mathcal{I}$ belonging to the layer $L_j$. Let $\mathcal{N}$ and $\mathsf{OPT}$ be two piercing sets for $\mathcal{I}$ returned by our algorithm and an offline optimal, respectively, for the input sequence $\mathcal{I}$. Let $\mathcal{N}_j$ be the piercing sets returned by our algorithm for $\mathcal{I}_j$. Let $p$ be any piercing point of an offline optimal $\mathsf{OPT}$. Let $\mathcal{I}_p \subseteq \mathcal{I}$ be the set of input fat objects pierced by the point $p$. For each $j \in [2\lfloor \log M \rfloor + 1] \cup \{0\}$, let $\mathcal{I}_{p,j} = \mathcal{I}_p \cap \mathcal{I}_j$. Let $\mathcal{N}_p$ be the set of piercing points placed by our algorithm to pierce all the fat objects in $\mathcal{I}_p$. For each $j \in [2\lfloor \log M \rfloor + 1] \cup \{0\}$, let $\mathcal{N}_{p,j} = \mathcal{N}_p \cap \mathcal{N}_j$ be the set of piercing points explicitly placed by our algorithm to hit hypercubes in $\mathcal{I}_{p,j}$. It is easy to see that $\mathcal{N} = \cup_{p \in \mathsf{OPT}} \mathcal{N}_p = \cup_{p \in \mathsf{OPT}} \left( \cup_{j=0}^{\lfloor \log M \rfloor} \mathcal{N}_{p,j} \right)$. Therefore, the competitive ratio of our algorithm is upper bounded by $\max_{p \in \mathsf{OPT}} (2\lfloor \log M \rfloor + 1) \times \max_j |\mathcal{N}_{p,j}|$.

Let $c$ be the center of an object $\sigma \in \mathcal{I}_{p,j}$. To hit $\sigma$, our algorithm adds a point $r \in \Pi_d^j$ such that $d(r, c) \leq \frac{\ell_j}{2}$ (due to Lemma 5). Since $c$ is the center of $\sigma \in \mathcal{I}_{p,j}$ having a width strictly less than $u_j$ and $p \in \sigma$, we have $d(c, p) < \frac{u_j}{\alpha}$. Now, using triangle inequality, we have $d(r, p) \leq d(r, c) + d(c, p)$. Consequently, we have $d(r, p) \leq \frac{u_j}{\alpha} + \frac{\ell_j}{2}$. Hence, an open hypercube $H_j$ of side length $\frac{2u_j}{\alpha} + \ell_j$, centered at $p$, contains all points in $\mathcal{N}_{p,j}$. Notice that $u_j \geq \frac{4}{3} \ell_j$. As a result, $H_j$ is open hypercube of side length $u_j(\frac{2}{\alpha} + \frac{3}{4})$.

**Observation 1.** *Let $\sigma$ be a hypercube with side lengths between $\ell\beta$ and $r\beta$, where $\ell$, $r$, and $\beta$ are positive real numbers such that $\ell < r$. Then, the object $\sigma$ contains at least $\lfloor \ell \rfloor^d$ and at most $\lfloor r+1 \rfloor^d$ points from $(\beta\mathbb{Z})^d$.*

Due to Observation 1, any open hypercube of side length $2^{i+1}(\frac{2}{\alpha} + 1)$ contains at most $\lfloor \frac{2}{\alpha} + \frac{7}{8} \rfloor^d$ points from $\Pi_d^j$. Thus, we have $|\mathcal{N}_{p,j}| \leq \lfloor \frac{2}{\alpha} + \frac{7}{8} \rfloor^d$. Recall that $|\mathcal{N}_p| = \cup_{i=0}^{\lfloor \log m \rfloor} |\mathcal{N}_{p,i}|$. Thus, we have $|\mathcal{N}_p| \leq \lfloor \frac{2}{\alpha} + \frac{7}{8} \rfloor^d (\lfloor 2 \log M \rfloor + 1)$.

## C    PSEUDO-CODES

---

**Algorithm 1** Algorithm ALGO-INTERVAL for Construction of Online $\varepsilon$-Net $\mathcal{N}$ for arbitrary intervals

---

1: Initialize net $\mathcal{N} = \emptyset$
2: **while** new interval $\sigma$ arrives **do**
3:    **if** $|\sigma \cap \mathcal{X}| < \varepsilon |\mathcal{X}|$ **then**
4:        Ignore $\sigma$.
5:    **else**
6:        **if** $\sigma$ is already hit by $N$ **then**
7:            Ignore $\sigma$.
8:        **else**
9:            Sort the points in $\sigma \cap \mathcal{X}$ as $p_1, p_2, \ldots, p_{|\sigma \cap \mathcal{X}|}$.
10:           Hit $\sigma$ with points indexed by $\lfloor \frac{|\sigma \cap \mathcal{X}|}{2} \rfloor$ and $\lceil \frac{|\sigma \cap \mathcal{X}|}{2} \rceil$.
11:           Add these points to $N$.
12:       **end if**
13:   **end if**
14: **end while**
15: **Return** $\mathcal{N}$

---

**Algorithm 2** Construction of Online $\varepsilon$-Net $N$ for axis-aligned rectangles

---

1: Fix a random sample $\mathcal{P} \subseteq \mathcal{X}$
2: Construct a balanced binary tree $\mathcal{T}$ over $\mathcal{P}$
3: For each node $v$, construct the set of maximal open $\mathcal{P}_v$-unhit rectangles $\mathcal{M}_v$
4: For each rectangle $M \in \mathcal{M}_v$ define, $w_M = \frac{s|M \cap \mathcal{X}|}{n}$.
5: Construct the safety-net $N_M$ for each $M \in \mathcal{M}_v$
6: Initialize empty set: $SN = \emptyset$
7: **while** an input $\varepsilon$-heavy rectangle $\sigma$ introduced **do**
8:    **if** $\sigma \cap \mathcal{P} = \emptyset$ **then**
9:        **if** $\sigma \cap SN = \emptyset$ **then**
10:           Find the highest node $v$ such that $l_v$ intersects $\sigma$.
11:           Identify a subrectangle $\sigma' \subseteq \sigma$ s.t $|\sigma'| \leq \frac{\varepsilon n}{2}$.
12:           Extend $\sigma'$ to form a $\mathcal{P}$-unhit rectangle $M \in \mathcal{M}_v$.
13:           Add points from safety-net $N_M$ to $SN$.
14:       **end if**
15:   **end if**
16: **end while**
17: **return** the final online net $N = \mathcal{P} \cup SN$.

---

