# OpenReview forum: "ONLINE EPSILON NET & PIERCING SET FOR GEOMETRIC CONCEPTS"
_ICLR.cc/2025/Conference — ICLR 2025 Poster_

### Official Review · Reviewer_42wh · 2024-10-31

**Soundness:** 3
**Presentation:** 3
**Contribution:** 3
**Rating:** 5
**Confidence:** 2

**Summary:**

This paper addresses the online ε-net and online piercing set problems for geometric concepts in computational geometry.  Specifically, it proposes  randomized algorithms for axis-aligned boxes in two- and three-dimensional spaces, achieving near-optimal competitive ratios, and  online algorithms for the piercing set problem  for more complex structures like ellipsoids and axis-aligned rectangles, presenting competitive ratios.

**Strengths:**

1. Novelty in Online Setup: The transition from offline to online algorithms for ε-nets and piercing sets addresses a gap in existing research. This online focus adds significant value to applications in dynamic learning environments.
2. Comprehensive Theoretical Analysis: Each algorithm is rigorously analyzed for its competitive ratio, demonstrating tight bounds for several common geometric configurations.

**Weaknesses:**

1. The competitive ratios for higher dimensions grow rapidly, limiting the algorithms’ effectiveness in high-dimensional applications (e.g., for dimensions $d \ge 4$). This can restrict practical use in real-world, high-dimensional datasets.
2. The results are mainly for regular, axis-aligned geometric objects (boxes, ellipsoids), yet real-world applications often involve irregular or rotated shapes. Without additional adaptability, the proposed algorithms would struggle in cases where piercing sets are needed for non-standard objects.
3. In the randomized approach for axis-aligned rectangles, a binary search tree is constructed over a subset of points, dividing intervals repeatedly until only ε-heavy regions remain. While theoretically sound, constructing and updating such a binary tree might be impractical in real-time applications due to overhead. A practical complexity analysis or alternative for dynamic binary tree maintenance would make the algorithm more viable.
4. The paper is heavily theoretical, with limited discussion of empirical validation. Experiments comparing the proposed algorithms' performance to existing methods would strengthen the practical implications.
5. Overall, I think this is a good paper, but might not be quite suitable for ICLR. Maybe the computational geometry or theory conferences, like SoCG, are more appropriate.

**Questions:**

1. While the theoretical competitive ratios are well-established, there is little discussion on the real-world computational cost. Does the approach become impractical under frequent updates, and if so, what modifications could reduce the computational load?
2. The paper does not provide a sensitivity analysis on the effect of different random sample sizes P in Section 3.2, which could significantly affect performance. Without such an analysis, it’s unclear whether the algorithm can reliably achieve near-optimal competitive ratios across different datasets and problem scales. How robust is the algorithm for online ε-nets under varying sample sizes?

---

> ### Author Response · Authors · 2024-11-26
> **Official Comments by the Authors**
>
> We thank you for your valuable time and feedback. Here are our responses to your concerns.
>
>
> 1. **While the theoretical competitive ratios are well-established, there is little discussion on the real-world computational cost. Does the approach become impractical under frequent updates, and if so, what modifications could reduce the computational load?**
>    **Response:** Since our paper largely focused on the online aspects of the problem, we didn't include the discussion on the computational cost which we felt a bit out of scope. However, in the revised manuscript (see Section 3), we have addressed the computational cost by analyzing the update time of the algorithm. Specifically, we construct a balanced binary tree $T$ over the points of $P$ in $O(n \log \frac{1}{\varepsilon})$ time. The construction of the sets $M_v$ for all nodes $v$ over all level $i$ takes $O(\frac{1}{\varepsilon^2})$ time. Additionally, the construction of safety-nets is linear in time. Consequently, the overall running time complexity of the algorithm never goes beyond $\max\{O(n \log \frac{1}{\varepsilon}),O(1/\epsilon^2)\}$, ensuring that the update time at each step does not exceed $\max\{O(n \log \frac{1}{\varepsilon}),O(1/\epsilon^2)\}$.
>
> 2. **The paper does not provide a sensitivity analysis on the effect of different random sample sizes $P$ in Section 3.2, which could significantly affect performance. Without such an analysis, it’s unclear whether the algorithm can reliably achieve near-optimal competitive ratios across different datasets and problem scales. How robust is the algorithm for online $\varepsilon$-nets under varying sample sizes?**
>    **Response:** The primary focus of this paper is to maintain a feasible online $\varepsilon$-net, for which the chosen sample size is effective and yields desirable results. Altering the size of the random sample would significantly impact the competitive ratio of the online algorithm. However, we have established a lower bound of $\Omega(\log \frac{1}{\varepsilon})$ for the problem (Theorem~2), and our online algorithm achieves a competitive ratio of $O(\log \frac{1}{\varepsilon})$ for online $\varepsilon$-nets of rectangles. Therefore, the algorithm is asymptotically optimal within this framework.

---

> > ### Comment · Reviewer_42wh · 2024-11-28
> > **Feedback**
> >
> > I appreciate the authors for the efforts spent in rebuttal. I think this is a good paper providing several novel results regarding eps-net. My only concern is that the topic is somewhat different with the scope of ICLR (and also a little restricted to specific geometric concepts). At this moment, I will keep my score, and am willing to discuss with other reviewers and AC in the later discussion phase, to see whether I need to modify the score.

---

### Official Review · Reviewer_RbqL · 2024-11-02

**Soundness:** 3
**Presentation:** 2
**Contribution:** 3
**Rating:** 6
**Confidence:** 2

**Summary:**

The paper studies the online $\epsilon$-net problem and the online piercing set problem for geometrical objects.

For the online $\epsilon$-net problem, the paper considers the range space of intervals in $\mathbb{R}^1$ and axis-aligned retangles in $\mathbb{R}^2$ and $\mathbb{R}^3$. For intervals, the paper gives an algorithm with asymptotically tight competitive ratio $O(\log(1/\epsilon))$. For axis-aligned rectangles, the paper gives an algorithm with nearly asymptotically tight competitive ratios ($O(\log(1/\epsilon))$ in $\mathbb{R}^2$ and $O(\log^3(1/\epsilon))$ in $\mathbb{R}^3$).

For the online piercing set problem, the paper considers the range spae of axis-aligned boxes, ellipsoids, and fat objects in $\mathbb{R}^d$ for general $d$. For axis-aligned boxes and ellipsoids, the paper gives an algorithm with an asymptotically tight competitive ratio $O(\log(M))$, where $M$ is the length range of the geometrical objects. For $\alpha$-fat objects, the paper gives an algorithm slightly improving the existing $ O\left(\left(\frac{2}{\alpha}+2\right)^d \log M\right)$ competitive ratio to $O\left(\left(\frac{2}{\alpha}+\frac{7}{8}\right)^d \log M\right)$ for $\alpha \in [1/2,1]$

**Strengths:**

The paper studies fundamental problems (online eps-net and online piercing set). The problem settings considered(range spaces of intervals, rectangles, ellipsoids, and fat-objects) are interesting and natural. The paper makes good contributions on these problems. In some problems, the paper achieves asymptotically optimal competitive ratio.

Overall, I believe the paper makes good contributions to a natural problem. Its contributions are worthy to publish.

**Weaknesses:**

The main weakness lies in the writing of the paper. The writing lacks rigor and causes distraction. Many concepts are not clearly defined. Some concepts are not clearly utilized. The connection between the existing results and this paper's result is not clearly stated. I have detailed these issues scifically in the Questions section below.

Overall, I believe the paper has the potential to be decent-to-good, provided that its writing is polished.

**Questions:**

1. The paper begins with an extensive discussion on VC-dimension; however, it does not utilize this concept in any of the subsequent analyses. All results depend solely on the Euclidean dimension, making the discussion of VC-dimension somewhat distracting for the reader. I recommend either removing this discusstion entirely or clearly articulating the relationship between the results and VC-dimension.

2. The connection between the paper's results and existing literature is not sufficiently clarified.

2.1. The authors claim that their results are asymptotically optimal. Could you specify which variable you are considering for the asymptotic limit?

2.2. The authors reference several existing lower bounds, some expressed in the forms $\log(n)$ and $\log(m)$. These bounds do not appear directly comparable to the results presented in this paper. Please elaborate on the relationships, similarities, and differences between these bounds and yours.

2.3. In line 132, the authors introduce a bound of the form $\log(M)$ without defining $M$, immediately following the introduction of the variable $m$. This lack of definition creates confusion, particularly regarding the similarity between uppercase $M$ and lowercase $m$. I suggest defining $M$ and potentially selecting a different variable to avoid this ambiguity.

3. Several terms and concepts lack formal definitions:

3.1. I recommend that the authors formally define $\epsilon$-nets in Section 2.

3.2. What is the definition of the hitting set problem? The authors frequently mention it and compare it to the piercing set problem without providing a clear definition. Since the paper does not focus on the hitting set problem, I suggest removing this discussion to maintain focus and coherence.

3.3. In line 42, the definition of a range space is missing.

3.4. The dual range space is introduced in line 75, but its definition is not provided until the following page.

3.5. The parameter $\alpha$ related to an object is not defined until later in the text (line 93).

3.6. In line 130 (and many other places ao, line 130 is just an example), the authors use the variables $P$ and $X$ interchangeably without defining $P$.

---

> ### Author Response · Authors · 2024-11-26
> **Official Comments by the Authors**
>
> Thank you for your feedback. We've reorganized the structure and notations for clarity. Below are our responses to your concerns.
>
> 1. **The paper begins with an $\cdots$ and VC-dimension.**
>
>     **Response:** The seminal result of Haussler and Welzl from 1987 states that it's possible to construct small size $\varepsilon$-net for set systems with bounded VC dimension. Moreover, a simple mechanism (e.g., random sampling) yields such guarantees. Unfortunately, we can't tackle objects with large VC dimensions. One simple example is when the elements are points in convex position, and ranges are convex subsets. For such a set system, there is a small $\varepsilon$-net. Hence, we need to critically rely on bounded VC dimension. Otherwise, any kind of sampling mechanism is not useful. In fact, it is somewhat easy to come up with strong lower bounds for set systems with finite but large VC dimensions.
>
>      On the other hand, the Euclidean dimension affects the union complexity of the objects. This, in turn, affects the construction of nets. We have addressed this issue in the manuscript (see Remark 1). For very restricted families of ranges, e.g., balls in $\mathbb{R}^d$, such connections could be established. However, already in $\mathbb{R}^2$, for simple range spaces (e.g., convex families), no such connection can be established. Generally, we are not aware of any connections between the Euclidean dimension and the VC dimension, and while they are not necessarily orthogonal, they are not directly related notions.
>
> 2. **The connection between the paper's results and existing literature is not sufficiently clarified.**
>
>     2.1. **The authors claim $\cdots$ for the asymptotic limit?**
>
>      **Response:** Thanks for pointing it out. We have added a proof of the optimal lower bound to avoid such confusion. In particular, we show that for online $\varepsilon$-net of intervals, any online algorithm has a competitive ratio of at least $\Omega(\log \frac{1}{\epsilon})$ (see Theorem~2 and the proof).
>
>      2.2. **The authors reference several existing $\cdots$ these bounds and yours.**
>
>      **Response:** Thank you for pointing this out. We have modified the related work section for better understanding and readability of the manuscript. Also, we have added a proof of the lower bound (Theorem~2) to avoid such confusion. Moreover, we have fixed the notational inconsistencies.
>
>      2.3. **In line 132, the authors introduce $\cdots$ a different variable to avoid this ambiguity.**
>
>      **Response:** Thank you for pointing this out. We have defined $M$ before using it. Here, $M$ is the ratio between the radius of the smallest ball containing the object and the radius of the largest ball contained inside the object. We have used $m$ to denote the cardinality of the set $R$.
>
> 3. **Several terms and concepts lack formal definitions:**
>
>     **Response:** Thank you for pointing this out. We have tackled all the mentioned comments in the manuscript.
>
>     3.1. **I recommend that the authors formally define $\varepsilon$-nets in Section 2.**
>
>       **Response:** We have formally defined $\epsilon$-nets in Section 2.
>
>     3.2. **What is the definition of the hitting set problem? The authors frequently mention it and compare it to the piercing set problem without providing a clear definition. Since the paper does not focus on the hitting set problem, I suggest removing this discussion to maintain focus and coherence.**
>
>       **Response:** Note that if we will consider the value of $\varepsilon$ to be $\frac{1}{\epsilon}$, then the $\varepsilon$-net problem becomes the hitting set problem. Hence, both of the problems are co-related. From a technical point of view as well, the algorithm for set cover and its dual hitting set for geometric families is typically via $\varepsilon$-net. Piercing on the other hand is simply the continuous version of where the entire space is the universe. For the sake of completeness, we have included the definition of the hitting set problem in revised manuscript (see Section~2).
>
>     3.3. **In line 42, the definition of a range space is missing.**
>
>       **Response:** We have included the definition in Section~2.
>
>     3.4. **The dual range space is introduced in line 75, but its definition is not provided until the following page.**
>
>       **Response:** We have included the definition in Section~1.
>
>     3.5. **The parameter $\alpha$ related to an object is not defined until later in the text (line 93).**
>
>       **Response:** We have included the definition in Section~1.
>
>     3.6. **In line 130 (and many other places, e.g., line 130 is just an example), the authors use the variables $P$ and $X$ interchangeably without defining $P$.**
>
>       **Response:** Thank you for pointing this out, in the revised manuscript only $X$ is used to denote the ground set elements (universal set). Now, $P$ is only used to denote the random sample from the ground set $X$ in Section~3.

---

> > ### Comment · Reviewer_RbqL · 2024-11-27
> > **Response to Authors**
> >
> > Dear Authors,
> >
> > Thanks for your detailed response. Your response addressess many of my concerns.
> >
> > However, I still strongly recommed the authors define concepts **before or immediately after** mentioning it, rather than pages later. My concerns 3.3, 3.4, 3.5 are all about this issue. In the rebuttal, you emphasize that these concepts are defined in the paper. I know that they are defined in the paper. My suggestion is about **where they are defined**. I strongly suggest the authors define concepts  **before or immediately after** mentioning it.
> >
> > This issue doesn't affect my score. Regardless of whether you choose to revise the manuscript to address this concern, I will keep my score as 6, given the scope of the paper (mainly about specific geometric objects). That said, please consider my suggestions.

---

> ### Author Response · Authors · 2024-11-28
>
> Regarding your concerns 3.3, 3.4, 3.5, we defined concepts before or immediately after mentioning them.

---

### Official Review · Reviewer_UMQk · 2024-11-04

**Soundness:** 4
**Presentation:** 3
**Contribution:** 3
**Rating:** 6
**Confidence:** 2

**Summary:**

The paper tackles the problems of online $\epsilon$-nets and online piercing sets for geometric concepts, specifically focusing on bounded VC-dimension and various types of geometric shapes such as intervals, axis-aligned boxes, and ellipsoids. The authors introduce several new algorithms that achieve optimal or near-optimal competitive ratios for these problems, marking the first known results in several of these areas.

**Strengths:**

1. This paper is one of the first to tackle online $\epsilon$-net and piercing set problems with a formalized approach, filling a noticeable gap in the existing literature.
2. The work provides clear proofs and establishes bounds on the competitive ratios of the proposed algorithms.

**Weaknesses:**

1. This paper mainly focuses on lower-dimensional cases (one-dimensional intervals and boxes in up to three dimensions). Many practical problems exist in higher dimensions, and the algorithms’ performance and applicability might not extend well to such cases.

**Questions:**

1. Can the authors extend their algorithms to work efficiently in higher dimensions, and if so, what are the expected challenges and potential solutions?
2. It would be beneficial if the authors could compare the performance of their online algorithms against existing offline methods to highlight the trade-offs and benefits more clearly.

---

> ### Author Response · Authors · 2024-11-26
> **Official Comments by the Authors**
>
> We thank the reviewer for their valuable feedback and comments, which helped us improve the quality of the manuscript. Here are our responses to your concerns.
>
> - **Can the authors extend their algorithms to work efficiently in higher dimensions, and if so, what are the expected challenges and potential solutions?**
>
>   **Response:** We would like to highlight a prospective challenge for extending the theoretical arguments in higher dimensions. Please note that, for dimensions $d \geq 4$, the number of maximal $\mathcal{P}$-unhit open orthants within each octant containing $k$ points from $\mathcal{P}$ may no longer be linear in $k$. In fact, it can grow as $\Theta(k^{\lfloor d/2 \rfloor})$, which is at least quadratic for $d \geq 4$ (see [3]). This contrasts with instances in $\mathbb{R}^2$ or $\mathbb{R}^3$, where the number of such maximal unhit boxes is linear in $k$, allowing us to efficiently bound the net size, which results in a small net size and a favorable competitive ratio.
>
>   However, due to the potentially non-linear growth in higher dimensions, it is unclear whether the tree construction algorithm used to find a small $\varepsilon$-net will yield similarly efficient results for $d \geq 4$. Moreover, the so-called *Union Complexity* (see [4]) landscapes for the objects in high dimensions are not well understood even in the offline scenarios. This stands as a major bottleneck to obtain strong lower bounds for the problem, which in turn, affects any online solution as well. We had mentioned these points in the form of a remark in the paper (see Remark 1 in Section 3).
>
>   Although the algorithms readily generalize to any dimension and indeed, through experimental evaluations, we have noticed the efficiency of our methods for instances in high dimensions. Since we couldn't prove theoretical guarantees (see discussion above), we have omitted these evaluations, which we would be happy to include in revised versions.
>
> - **It would be beneficial if the authors could compare the performance of their online algorithms against existing offline methods to highlight the trade-offs and benefits more clearly.**
>
>   **Response:** The competitive ratio of our algorithms is based on the comparison with an offline optimum solution, where we obtained logarithmic lower bounds in a very simple one-dimensional setup. For the upper bounds, in most cases, we asymptotically match these lower bounds.
>
>   However, we can't provide theoretical guarantees for a more general set system in very high dimensions due to some extremely challenging bottlenecks (please see the response to the last point of Reviewer 1). However, our algorithmic framework works in any dimension, and through experimental evaluation, we observed that the algorithms work well in practice.
>
>
> **Bibliography**
>
> [3] Haim Kaplan, Natan Rubin, Micha Sharir, and Elad Verbin. *Efficient colored orthogonal range counting*.
> SIAM Journal on Computing, 38(3):982–1011, 2008.
>
> [4] Kasturi R. Varadarajan: *Epsilon nets and union complexity*. SCG 2009: 11-16

---

> > ### Comment · Reviewer_UMQk · 2024-11-28
> >
> > I thank the authors for the responses. After reading other reviews and responses, I will keep my score.

---

### Official Review · Reviewer_bAWb · 2024-11-05

**Soundness:** 3
**Presentation:** 2
**Contribution:** 3
**Rating:** 6
**Confidence:** 2

**Summary:**

This is a theoretical paper that gives algorithms to construct epsilon nets and piercing sets for geometric objects that appear in an online setting. The size of the nets and piercing sets are tight in terms of the competitive ratio i.e. the largest ratio of the size of the net returned by the algorithm to the optimal size of the net at any point in the online setting.

**Strengths:**

1) Epsilon nets are very important in the field of computational geometry and learning theory. It appears that very less work is done regarding construction of epsilon nets/piercing nets when the objects appear in an online manner. Hence the paper will be of interest to the community.

2) I tried to check the proofs in the main part of the paper to the best of my ability and the paper appears to be sound.

3) The techniques/insights described here might be useful to construct epsilon nets for other objects also. I appreciate the use of figures to convey certain ideas.

**Weaknesses:**

1) The writing and the presentation of the paper is very weak. Parts of the paper are very abstract. Certain words and concepts appear earlier and are explained later and that too in abstract manner. The writing is repetitive also. Here I give few specific examples:

i) In the Notation and Preliminaries section there is hardly any difference in the definition of online epsilon net and online piercing set.

ii) Terms like ranges, range space are used in the introduction section without properly defining them.

iii) You can give simple examples for what can be $\mathcal{X}, \mathcal {R}$ etc. in the introduction itself. The final results can be abstract but at least some examples might help understand the terminology better.

iv) It would be better to bring the pseudo code of the algorithms in the main paper

The writing and presentation need to be improved a lot.  In the current state the paper will not be accessible to a broad audience.

2) It is not clear from the paper what are the technical challenges faced to maintain an epsilon net in the online setting. Please clarify as it will help highlight the novelty as to why the techniques of constructing the epsilon net differ from the offline case. For this you can add a small paragraph has to how epsilon nets can be constructed for similar object in the offline setting and why it is difficult to modify such algorithms for the online setting.

**Questions:**

See Weaknesses and address them

---

> ### Author Response · Authors · 2024-11-26
> **Official Comments by the Authors**
>
> - **The writing and the presentation of the paper is very weak. Parts of the paper are very abstract. Certain words and concepts appear earlier and are explained later and that too in an abstract manner. The writing is repetitive also. Here I give a few specific examples:**
>
>   **Response:** We thank the reviewer for giving valuable feedback regarding the writing. We have significantly improved our writing by rearranging definitions and notations for clarity, and we believe it improved the exposition of the results. In what follows, we address the specific concerns expressed by the reviewer.
>
>   1. **In the *Notation and Preliminaries* section, there is hardly any difference in the definition of the online epsilon net and online piercing set.**
>
>      **Response:** We have included earlier the definitions for both problems explicitly for the sake of completeness. However, we are happy to remove the formal definition for the piercing set, for which an informal definition is already included in the manuscript (see Line 80).
>
>   2. **Terms like *ranges*, *range space* are used in the introduction section without properly defining them.**
>
>      **Response:** Thank you for pointing this out. Since the notion of ranges and range spaces are commonly used terminologies in Statistical Learning Theory, we didn't define them explicitly. However, for completeness, we have included the definitions in the current version (see Section 1).
>
>   3. **You can give simple examples for what can be $(\mathcal{X}, \mathcal{R})$ etc. in the introduction itself. The final results can be abstract but at least some examples might help understand the terminology better.**
>
>      **Response:** We have included an explicit example in the introduction (see Section 1).
>
>   4. **It would be better to bring the pseudo-code of the algorithms in the main paper.**
>
>      **Response:** Due to space constraints, we couldn't include the pseudo-codes inside the main body, which would have compelled us to keep most of the formal arguments in the Appendix. We would be happy to include the pseudo-codes in the revised version of the paper.
>
> - **The writing and presentation need to be improved a lot. In the current state, the paper will not be accessible to a broad audience.**
>
>   **Response:** We have modified the introduction with the appropriate references and examples. $\varepsilon$-nets and hitting sets are some of the basic concepts in Learning Theory. We believe the current version of the paper will connect to a broader set of audiences. Furthermore, we have modified the writing of the notation and preliminaries sections to improve the readability of the paper.
>
> - **It is not clear from the paper what are the technical challenges faced to maintain an epsilon net in the online setting. Please clarify as it will help highlight the novelty as to why the techniques of constructing the epsilon net differ from the offline case. For this, you can add a small paragraph on how epsilon nets can be constructed for similar objects in the offline setting and why it is difficult to modify such algorithms for the online setting.**
>
>   **Response:** One key and fundamental difference is that in the offline setup, the entire set of objects is known in advance, in contrast to the online setting where objects arrive one by one. Moreover, in the online setup, decisions are taken irrevocably, i.e., upon arrival of a new object at any step, the online algorithm needs to maintain the $\varepsilon$-net which can't be changed in subsequent steps. This marks a sharp contrast with the offline setup since the adversary could design a worst-case scenario, which is difficult for any online algorithm to combat.
>
>   Of particular interest, we want to highlight some technical challenges. For instance, one commonly used analysis technique in the offline $\varepsilon$-net is to use the so-called *packing lemma* and *shallow packing lemma*, which informally state that given a set system with bounded VC-dimension, for every distinct pair of subsets of ranges, if the cardinality of the symmetric difference is bounded by some constant, then the cardinality of the range set is polynomially bounded by the number of elements (see~[1,2]) ]for a formal description and the proof of the offline $\varepsilon$-net). However, in the online case, since the ranges are not known in advance, the symmetric difference argument cannot be used appropriately.
>
> **Bibliography**
>
> [1] Jiří Matoušek. *Reporting points in halfspaces*. Computational Geometry, 2(3):169–186, 1992.
>
>
> [2] Nabil H. Mustafa. *Computing optimal epsilon-nets is as easy as finding an unhit set*. In Christel Baier,
> Ioannis Chatzigiannakis, Paola Flocchini, and Stefano Leonardi, editors, 46th International Colloquium
> on Automata, Languages, and Programming, ICALP 2019, July 9-12, 2019, Patras, Greece, volume 132
> of LIPIcs, pages 87:1–87:12. Schloss Dagstuhl - Leibniz-Zentrum f¨ur Informatik, 2019.

---

> > ### Comment · Reviewer_bAWb · 2024-11-28
> > **Thanks for the reply**
> >
> > Thanks for the rebuttal. I have read the rebuttal and the other reviews. I agree with Reviewer RbqL  regarding the writing , definition placement and also scope. I will keep my score.

---

### Meta-Review · Area_Chair_yEsM · 2024-12-11

**Metareview:**

This work presents new algorithms for online eps-net and piercing set of certain geometric concepts. While the concept classes studied in this paper are quite limited, the theory still advances the literature via establishing almost matching lower and upper bounds and may inspire future research. Based on the reviews and the AC's own reading, the AC believes that the results are significant enough.

**Additional Comments On Reviewer Discussion:**

This is a technically heavy work and all reviewers have quite low confidence. Most questions are about clarity, which authors had addressed.

It should be noted that while the average rating is slightly below '6', all reviewers have low confidence. Thus, the AC read all reviews, discussions, and the manuscript carefully. As an expert in learning theory, the AC considers the paper well written (though likely not friendly to non-expert) and the contribution significant.

---

### Decision · Program_Chairs · 2025-01-22

Accept (Poster)